# The ribosome-associated complex RAC serves in a relay that directs nascent chains to Ssb

Ying Zhang[1,3], Genís Valentín Gesé [2,3], Charlotte Conz[1], Karine Lapouge [2], Jürgen Kopp[2], Tina Wölfle[1], Sabine Rospert [1✉] & Irmgard Sinning [2✉]

The conserved ribosome-associated complex (RAC) consisting of Zuo1 (Hsp40) and Ssz1 (non-canonical Hsp70) acts together with the ribosome-bound Hsp70 chaperone Ssb in de novo protein folding at the ribosomal tunnel exit. Current models suggest that the function of Ssz1 is confined to the support of Zuo1, however, it is not known whether RAC by itself serves as a chaperone for nascent chains. Here we show that, via its rudimentary substrate binding domain (SBD), Ssz1 directly binds to emerging nascent chains prior to Ssb. Structural and biochemical analyses identify a conserved LP-motif at the Zuo1 N-terminus forming a polyproline-II helix, which binds to the Ssz1-SBD as a pseudo-substrate. The LP-motif competes with nascent chain binding to the Ssz1-SBD and modulates nascent chain transfer. The combined data indicate that Ssz1 is an active chaperone optimized for transient, low-affinity substrate binding, which ensures the flux of nascent chains through RAC/Ssb.

[1] Institute of Biochemistry and Molecular Biology, ZBMZ, Faculty of Medicine, University of Freiburg, D-79104 Freiburg, Germany; BIOSS Centre for Biological Signaling Studies, University of Freiburg, D-79104 Freiburg, Germany. [2] Heidelberg University Biochemistry Center (BZH), INF 328, D-69120 Heidelberg, Germany. [3] These authors contributed equally: Ying Zhang, Genís Valentín Gesé. ✉email: sabine.rospert@biochemie.unifreiburg.de; irmi.sinning@bzh.uni-heidelberg.de

Ribosome-associated chaperones play a fundamental role in de novo protein folding in all kingdoms of life. They encounter nascent polypeptides when they emerge from the ribosomal exit tunnel and act before cytosolic chaperones become involved in folding as translation proceeds[1,2]. Eukaryotic cells possess a specialized ribosome-associated complex (RAC), which in yeast is a stable heterodimer formed by the Hsp70 homolog Ssz1 and the J-domain protein Zuo1[3,4] (Fig. 1a). RAC is the J-domain partner of the ribosome-associated Hsp70 homolog Ssb (encoded by two nearly identical isoforms SSB1 and SSB2). Ssb cotranslationally interacts with a variety of nascent chains when they emerge at the ribosomal tunnel exit[3,5–7]. Binding of Ssb to nascent chains depends on an intact Zuo1 J-domain and Ssz1[8]. Consistently, both RAC subunits are required to efficiently stimulate ATP hydrolysis by Ssb[9].

Ssb is a canonical Hsp70 protein comprising an N-terminal nucleotide binding domain (NBD), an interdomain linker, and a C-terminal substrate binding domain (SBD), composed of SBDβ and the SBDα lid domain[3,4,10]. Ssb contacts ribosomal proteins and rRNA segments, which localize around the tunnel exit[4,10] in proximity to RAC[11,12]. Zuo1 possesses a complex domain structure with a long α-helical middle domain (MD), which separates two major functional domains. The N-terminal part comprises an N-terminal domain (N), the J-domain (J) and the zuotin homology domain (ZHD). Zuo1 contacts the 60 S subunit via the ZHD close to Rpl31 (eL31) and H24 of the 25 S rRNA at the tunnel exit[4,11–13]. The Zuo1 C-terminal domain consists of a 4-helix bundle (HD)[14,15], which contacts the 40S subunit at the tip of h44 ES12[4,12]. Ssz1 significantly differs from canonical Hsp70 homologs. The Ssz1-NBD binds ATP, but does not hydrolyze ATP[9,14]. Our recent structure of the RAC core (consisting of Ssz1 and Zuo1N residues 19 to 60) shows that the Ssz1 interdomain linker, which is critical for allosteric regulation of canonical Hsp70s[16] has a unique structure and is detached from the NBD[17]. Moreover, Ssz1 lacks the C-terminal SBDα lid domain, but contains a complete SBDβ complemented by Zuo1N[17]. Thus, from all we know, Ssz1 cannot undergo the canonical Hsp70 cycle of structural rearrangements required for substrate binding and release[18]. Besides, in vitro crosslinking experiments, which readily reveal the interaction of Ssb with nascent chains, up to now did not reveal an interaction of Ssz1 with nascent chains[8]. Based on these observations the current model is that Ssz1 does not act as a typical Hsp70 chaperone. Instead, it may primarily play a structural role, supporting Zuo1 function as the J-domain partner of Ssb[9,19].

Here we investigate whether the RAC subunits interact with nascent chains. Our results indicate that both RAC subunits contact nascent chains and form a relay that transfers them from Ssz1 to Ssb. Structural analysis reveals that the Zuo1 N-terminus contains a conserved motif (LP-motif) which binds to the Ssz1-SBDβ in the same way as canonical Hsp70s bind their substrates. This motif competes with nascent chain binding, and modulates the interaction of the nascent chain with Zuo1 and Ssz1. Overall, we show that Ssz1 is an active chaperone with specific deviations from the canonical Hsp70 mechanism.

## Results

**Nascent chains contact Zuo1 and Ssz1 prior to Ssb**. As the structure of the RAC core shows the presence of a complete SBDβ[17], RAC should be able to bind substrates with low affinity besides its function as J-domain partner of Ssb. To test for the possibility, we generated ribosome nascent chain complexes (RNCs) and analyzed nascent chain contacts via a crosslinking approach using Saccharomyces cerevisiae (Sc) as a model system (Fig. 1b)[20]. As a substrate we employed the soluble, cytosolic

enzyme phosphoglycerate kinase (Pgk1), which is a well characterized nascent chain[20,21]. As expected, Ssb formed a weak crosslink to 50 residues nascent Pgk1 (Pgk1-50), while Ssb formed more efficient crosslinks with 100- or 150 residues nascent Pgk1, respectively[5,6]. Consistent with previous crosslinking analysis of nascent prepro α-factor[8], crosslinks between Zuo1 or Ssz1 and nascent Pgk1-50, or longer, were below the detection limit (Supplementary Fig. 1a).

We next tested the interaction of Ssb and the RAC subunits with nascent Pgk1-40 and Pgk1-45, respectively. Ssb did not form crosslink products with Pgk1-40 or Pgk1-45 (Fig. 1c). This is consistent with previous data indicating that the interaction with Ssb requires a minimal nascent chain length of about 50 residues[5–7]. However, Zuo1 and Ssz1 formed crosslinks with these short nascent chains (Fig. 1c). Zuo1 was most efficiently crosslinked to Pgk1-40, while Ssz1 was most efficiently crosslinked to Pgk1-45 (Fig. 1c). These crosslinks disappeared at the expense of the Ssb crosslink when the nascent chain reached a length of 50 residues (Fig. 1c). The data suggested that the nascent chain was transferred via a chain of interactions from Zuo1, to Ssz1, and finally to Ssb.

To further investigate the handover process, we tested whether RAC was in contact with longer nascent chains when Ssb was absent, i.e. when nascent chains were generated in a translation extract derived from Δssb1Δssb2 cells (Fig. 1d, Supplementary Fig. 1b). Indeed, when Ssb was absent, Zuo1 and Ssz1 formed efficient crosslinks to nascent chains of 50–150 residues (Fig. 1d Δssb and Supplementary Fig. 1b). Seemingly, Ssb prevented the interaction of Zuo1 and Ssz1 with long nascent chains, which is consistent with a direct and efficient nascent chain transfer from RAC to Ssb. To further corroborate this model, we used an Ssb mutant, which carries a deletion of its C-terminal 23 residues (Ssb-ΔC23), which does not stably interact with ribosomes (Supplementary Fig. 1c and 1d)[10,22]. When nascent chains were generated in a translation extract derived from Ssb-ΔC23 cells, no crosslink between Ssb-ΔC23 and Pgk1-100 was detected, but instead nascent Pgk1-100 remained in close contact with the RAC subunits (Fig. 1d). Thus, efficient crosslinking between the nascent chain and Ssb was dependent on the positioning of Ssb at the exit of the ribosomal tunnel. If Ssb was not bound to the ribosomal tunnel exit region, the nascent chain maintained its contacts with the RAC subunits.

We next asked if both subunits of RAC were required for the handover to ribosome-bound Ssb. To that end, we analyzed the interaction of Ssb1 with nascent Pgk1-100 (i) in the absence of both RAC subunits; (ii) in the presence of the N-terminally truncated Zuo1-ΔN49 mutant, which does not stably interact with Ssz1 leading to a situation in which Zuo1 is ribosome-bound, but Ssz1 is not[17] (Supplementary Fig. 1e, Zuo1-ΔN49); and (iii) in the presence of Zuo1, but absence of Ssz1 (Fig. 1e). Only in the presence of wild type RAC nascent Pgk1-100 formed an efficient crosslink to Ssb (Fig. 1e, XL-Ssb). When Ssz1 was absent or was not in a stable complex with Zuo1 the nascent chain remained in contact with Zuo1, even though Ssb was associated with ribosomes (Fig. 1e, XL-Zuo1; Supplementary Fig. 1f). The findings strongly suggested that Ssz1 was required for the handover from Zuo1 to ribosome-bound Ssb. This was further corroborated when nascent Pgk1-100 was generated in a translation extract derived from a Δssb1Δssb2 strain and purified Ssb1 was added prior to the translation reaction (Fig. 1f; Supplementary Fig. 1g,h). Consistent with the data shown above both RAC subunits formed crosslinks with the nascent chain in the absence (Fig. 1f, Δssb extract), but not in the presence of Ssb (Fig. 1f, wt). When purified Ssb (Supplementary Fig. 1g) was added to the Δssb1Δssb2 extract, crosslinking to the RAC subunits was strongly reduced and crosslinking of Ssb to nascent Pgk1-100 was restored (Fig. 1f,

Supplementary Fig. 1h). Taken together, these data reveal three novel functions of RAC. First, both RAC subunits are in close contact with nascent chains when these emerge at the tunnel exit. Second, the RAC subunits remain in contact with the nascent chain for a prolonged period of time, when Ssb is absent or is not bound to the ribosome. Third, the presence and proper positioning of Ssz1 as well as Ssb at the tunnel exit is required for the transfer of the growing nascent chain from Zuo1 to Ssb.

**Zuo1N extends the Ssz1-SBDβ.** Having established that Ssz1 and Zuo1 both interact with nascent chains, we aimed to understand

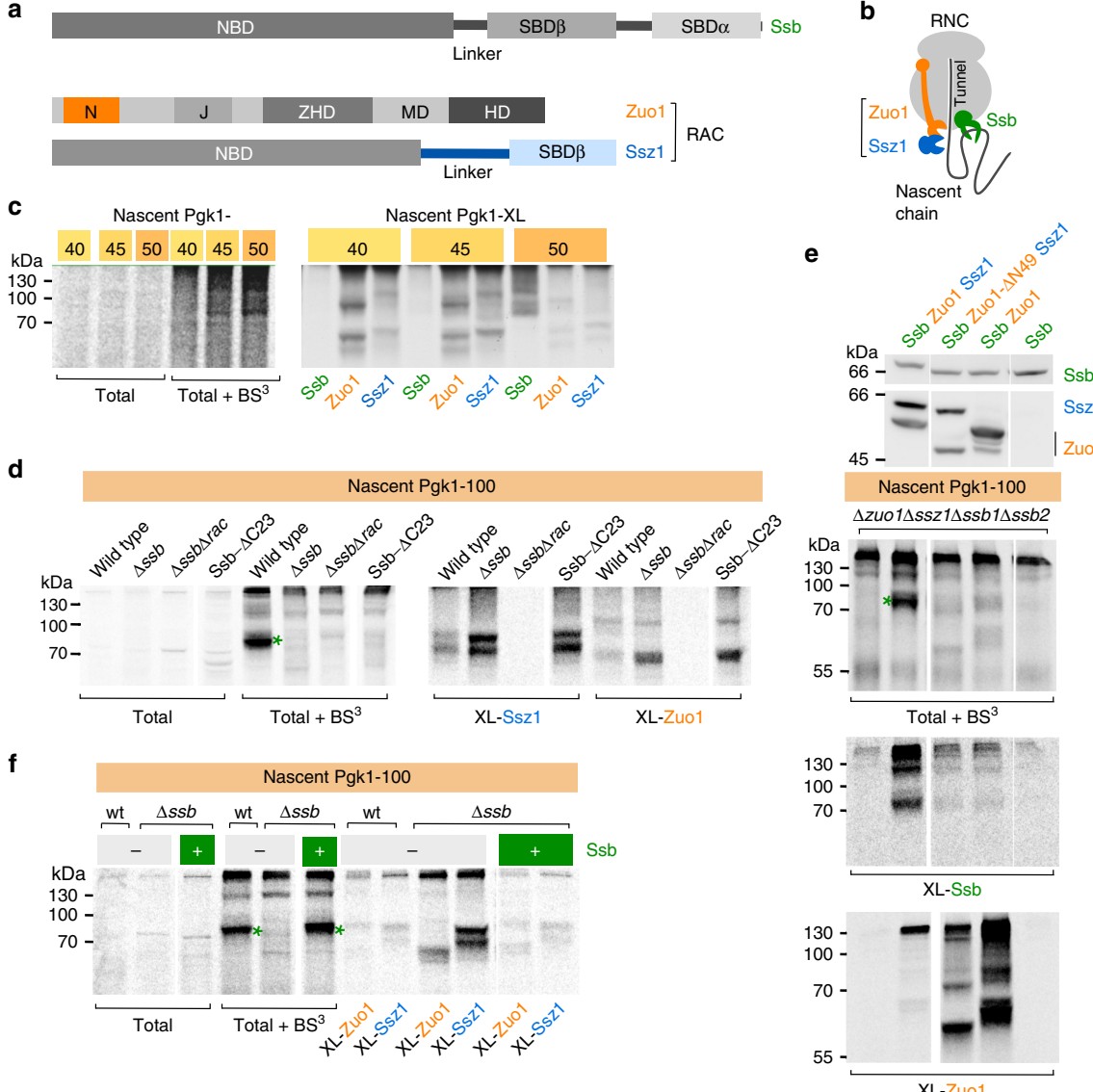

**Fig. 1 Zuo1 and Ssz1 contact nascent chains prior to Ssb. a** Domain structure of yeast Ssb, Zuo1, and Ssz1. Nucleotide binding domain (NBD), linker, substrate binding domain β (SBDβ), substrate binding domain α (SBDα), N-terminal domain (N), J-domain (J), Zuo1 homology domain (ZHD), middle domain (MD), and four-helix bundle (HD). **b** Experimental set up of crosslinking experiments. Isolated RNCs carrying [$^{35}$S]-labelled nascent chains are crosslinked to adjacent proteins using the homobifunctional amino-reactive crosslinker BS$^3$, spacer length 11.4 Å. Crosslink products between nascent chains and Ssb, Ssz1, or Zuo1 are then identified via immunoprecipitation under denaturing conditions using antibodies directed against Ssb, Ssz1, or Zuo1. For details see Methods. **c** Contacts of Ssb, Zuo1, and Ssz1 with short nascent chains. RNCs carrying Pgk1-40 (40 residues), -45 (45 residues), or -50 (50 residues) were generated in a wild type translation extract as described in (**b**). Crosslink products (Pgk1-XL) between Ssb and nascent Pgk1 (Ssb); Zuo1 and nascent Pgk1 (Zuo1); and Ssz1 and nascent Pgk1 (Ssz1). **d** Zuo1 and Ssz1 interact with nascent Pgk1-100 (100 residues) in the absence of ribosome-bound Ssb. The experiment was performed as described in **c** using RNCs from wild type, Δssb1Δssb2 (Δssb), Δssb1Δssb2Δzuo1Δssz1 (ΔssbΔrac), or Ssb1-ΔC23 translation extract. The green asterisk indicates the crosslink between Ssb and Pgk1-100 in the total (**d**–**f**). **e** Transfer of the nascent chain from Zuo1 to Ssb requires Ssz1. The experiment was performed as in (**c**) using RNCs from Δzuo1Δssz1Δssb1Δssb2 translation extract. Ribosome-free extracts (upper panel) obtained from wild type (Ssb, Zuo1, Ssz1), Δzuo1 expressing Zuo1-ΔN49 (Ssb, Zuo1-ΔN49, Ssz1), Δssz1 (Ssb, Zuo1), Δzuo1Δssz1 (Ssb) were added after completion of translation reactions, prior to crosslinking. Rebinding of Ssb to RNCs was analyzed as described in Supplementary Fig. 1f. **f** Ssb prevents crosslinking of Zuo1 and Ssz1 to long nascent chains. The experiment was performed as in (**c**) using RNCs from wild type (wt) or Δssb1Δssb2 (Δssb) translation extract. Purified Ssb1 (Supplementary Fig. 1g,h) was added prior to the translation reaction as indicated (+ Ssb). Source data for (**c**–**e**) are provided as a Source Data file.

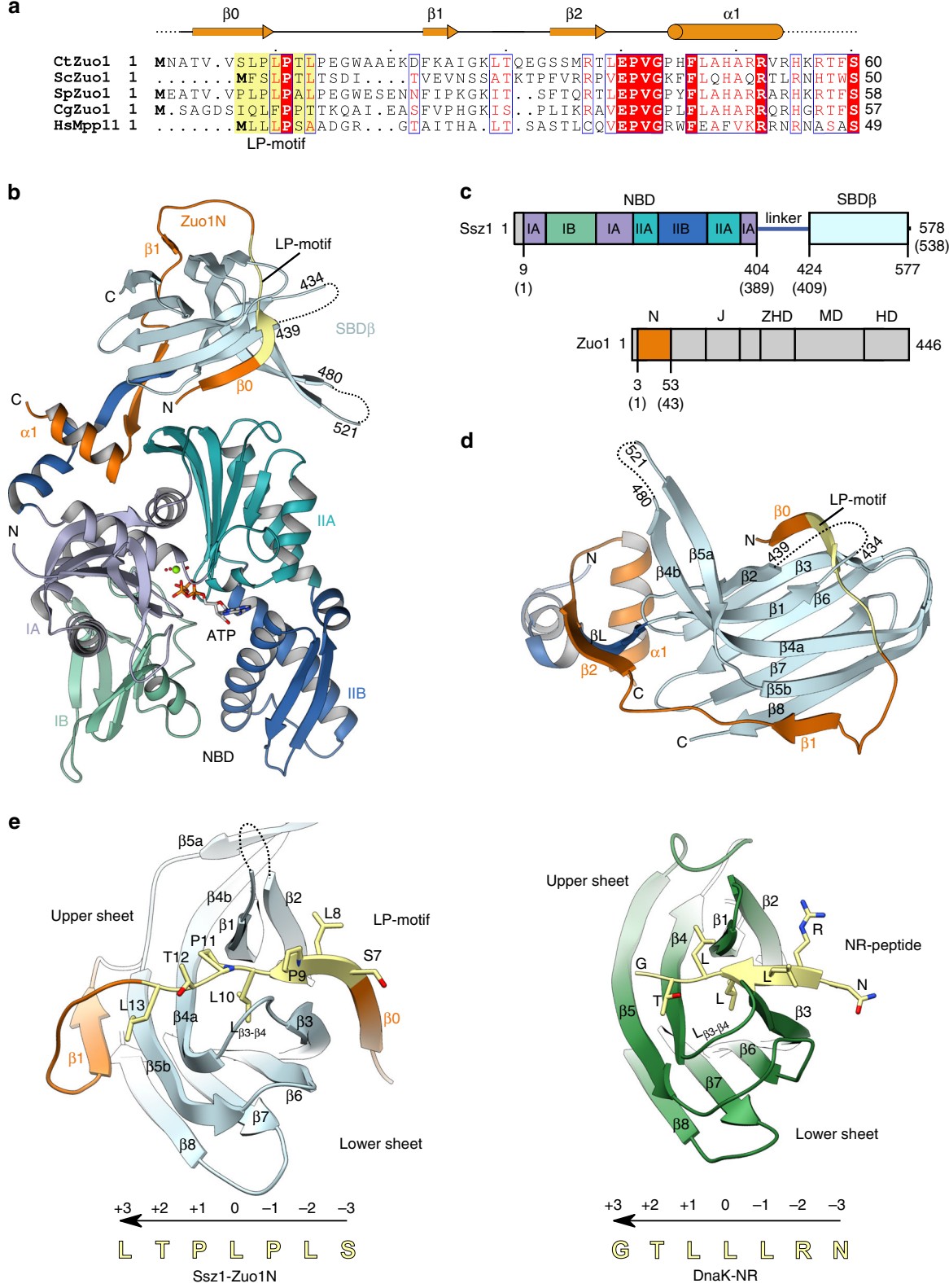

how RAC could accommodate a nascent chain. Previous structural data of the RAC core from *Chaetomium thermophilum* (*Ct*), consisting of full length *Ct*Ssz1 and *Ct*Zuo1N (residues 19–60), showed that it contains a complete Ssz1-SBD β-sandwich domain, which should allow for low-affinity substrate binding[17]. Complementation of the Ssz1-SBDβ by Zuo1N *in trans* creates a mixed RAC–SBD with a topology seen in canonical Hsp70s in the closed (ADP-bound) state. However, attempts to bind typical Hsp70 substrate peptides to RAC in vitro failed[17].

We now determined the structure of an N-terminally extended RAC core comprising *Ct*Ssz1 with the complete *Ct*Zuo1N (residues 1–60 of *Ct*Zuo1; Fig. 2a,b) at 2.5 Å resolution. The structure has been solved by molecular replacement using the previous RAC core as a search model (Fig. 2a–d and Table 1).

**Fig. 2 The structure of the Ssz1-Zuo1N complex reveals binding of a pseudo-substrate. a** Multiple sequence alignment of Zuo1N. The secondary structure derived from the crystal structure is depicted above. Ct, *Chaetomium thermophilum*; Sc, *Saccharomyces cerevisiae*; Sp, *Schizosaccharomyces pombe*; Cg, *Chaetomium globosum*; Hs, *Homo sapiens*. The alignment is color coded by conservation: strictly conserved residues are in a red box, similar residues are in red font. The LP-motif is highlighted in yellow. **b** Crystal structure of *Ct*Ssz1-Zuo1N in ribbon representation. The Ssz1-NBD, Ssz1-SBDβ and Zuo1N are colored as in (**c**). The LP-motif is highlighted in yellow. Disordered residues are indicated as dotted lines. The ATP, Mg[2+] and coordinated waters are shown in stick and sphere representation. **c** Domain architecture of Ssz1 and Zuo1. Domains present in the crystal structure are indicated by residue numbers for *Ct* (*Sc* numbers are in parentheses). Domain designation is as in Fig. 1a. **d** Zoomed-in view on the interaction of Ssz1-SBDβ with Zuo1N, showing how Zuo1N extends the SBDβ. Coloring is as in (**c**). **e** Comparison of substrate binding to Ssz1-SBDβ (left) and DnaK (right) ([23]; PDB 1DKZ). The SBDα of DnaK is not shown. Side chains of the LP-motif and NR-peptide are shown as stick representations. Sequences and direction (N- to C-terminus) of the LP-motif and NR-peptide are shown below the structural representations. Numbers indicate the position of each residue with respect to the leucine that occupies the center of the substrate binding pocket (position 0). Coloring of Ssz1 and Zuo1N is as in (**c**). DnaK is shown in green and the NR-peptide in yellow.

## Table 1 Data collection and refinement statistics (molecular replacement).

| | Ssz1-Zuo1N |
|---|---|
| **Data collection** | |
| Space group | $P\,1\,2_1\,1$ |
| Cell dimensions | |
| a, b, c (Å) | 52.0, 258.5, 53.0 |
| α, β, γ (°) | 90.0, 100.1, 90.0 |
| Resolution (Å) | 48.40–2.5 (2.59–2.5)[a] |
| Rmerge | 13.4 (70.6) |
| I/σ(I) | 9.6 (2.2) |
| CC$_{1/2}$ (%) | 99.2 (65.8) |
| Completeness (%) | 98.9 (96.8) |
| Redundancy | 6.0 (5.5) |
| **Refinement** | |
| Resolution (Å) | 48.40–2.5 (2.59–2.5) |
| No. reflections | 46791 (4582) |
| $R_{work}$ /$R_{free}$ (%) | 18.89 / 22.34 |
| No. atoms | |
| Protein | 8732 |
| Ions (Mg[2+]) | 2 |
| Ligands (ATP) | 62 |
| Water | 91 |
| B factors (Å$^2$) | |
| Protein | 53.71 |
| Ion (Mg[2+]) | 34.3 |
| Ligand (ATP) | 34.56 |
| Water | 36.3 |
| R.m.s. deviations | |
| Bond lengths (Å) | 0.004 |
| Bond angles (°) | 0.98 |

Each structure was determined from one crystal.
[a]Values in parentheses are for the highest-resolution shell.

Overall, the structure of Ssz1-Zuo1N[1–60] superimposes well with the previous structure[17] (Supplementary Fig. 2a,b). The Ssz1-NBD adopts the typical actin-like fold of Hsp70s, which is divided into two lobes (I and II), with ATP and Mg[2+] bound between them (Fig. 2b). The domain arrangement is as before, with the Ssz1 linker detached from the Ssz1-NBD creating a loophole for Zuo1N and with the conserved Zuo1 EPVG motif contacting the Ssz1-NBD at the linker binding site (Fig. 2b, Supplementary Fig. 2b).

However, the structure of the Ssz1-Zuo1N[1–60] complex reveals a number of novel, functionally important details. First, Ssz1-SBD β8 is now ordered and results in an Ssz1 β-sandwich composed of a 4-stranded upper (β2, β1, β4a, β5b), and a 4-stranded lower sheet (β3, β6, β7, β8) (Fig. 2d and Supplementary Fig. 2c,d) as characteristic for canonical Hsp70-SBDβs in the closed state[23,24]. In contrast to the previous structure (Ssz1-Zuo1N[19–60]) in which Ssz1 β8 was disordered and its position taken by Zuo1N β1[17],

Ssz1 β8 is now structured and aligns with β7 of the lower sheet. Second, the N-terminal extension of Zuo1N contributes two additional β strands (β0, β1) to Ssz1-SBDβ resulting in a β-sandwich with two mixed, 5-stranded β sheets. Zuo1N β0 complements the lower and β1 now the upper sheet of Ssz1-SBDβ, aligning with Ssz1-SBD β3 and β5b, respectively (Fig. 2d and Supplementary Fig. 2c,d). While this extension of both β-sheets by Zuo1 stabilizes the SBD, these differences also illustrate the plasticity of the SBD, which is underlined by the high B factors observed especially in the loop regions (Supplementary Fig. 3a,b). Notably, Zuo1N Phe23 inserts into a hydrophobic pocket between the upper and lower β-sheet of the Ssz1-SBD (Supplementary Fig. 3c,d). While this phenylalanine is not strictly conserved, the position is always occupied by a hydrophobic residue (e.g. *Sc* Val13) pointing to a structural role. In summary, the RAC core structure shows that Zuo1N embraces the Ssz1-SBDβ, resulting in an extended RAC–SBD.

**A proline-rich motif in Zuo1N binds to the RAC–SBD.** A third insight gained from the Ssz1-Zuo1N structure was that a conserved region (SLPLPTL, termed LP-motif, Fig. 2a) present in *Ct*Zuo1N between β0 and β1 binds to the RAC–SBD in a substrate-like manner (Fig. 2c–e). Canonical Hsp70s preferentially bind peptides containing hydrophobic residues at the central positions[23–25]. A detailed comparison of the RAC core with the DnaK-SBD structure with bound NRLLLTG peptide (NR-peptide, that serves as an Hsp70 model substrate)[23] shows the conservation of the SBDβ structures and their substrate binding modes (Fig. 2e, Supplementary Fig. 4a-d). The Zuo1 LP-motif and the NR-peptide overall establish similar interactions with their cognate SBDs. Zuo1 Leu10 occupies the hydrophobic center of the Ssz1 substrate binding pocket and corresponds to the central Leu (0 position) in the NRLLLTG peptide[23]. Residues Leu8, Pro9, Leu10 and Thr12 of the LP-motif establish H-bonds with backbone atoms of Ssz1-SBDβ strands β3, β1, β3 and β4, respectively, while the conserved Pro11 does not form a hydrogen bond (Supplementary Fig. 4b). The loop connecting β3 and β4 (L$_{β3-β4}$, disordered in the previous structure) is ordered due to stabilization by the LP-motif (Fig. 2e, Supplementary Fig. 4b,d). Interestingly, the position of the NR-peptide in DnaK[23,26] and other Hsp70s, like BiP and human cytosolic Hsp70[27,28], can be shifted by one residue due to crystal packing, which locates the third Leu to the central position (NRLLLTG), but maintains the hydrogen-bonding pattern. Notably, the Ssz1-bound LP-motif adopts an almost ideal poly-L-proline type II (PPII) helix, which resembles the binding mode of proline-rich antimicrobial peptides[26], while the DnaK-bound NR-peptide adopts a more extended conformation (Fig. 2e, Supplementary Fig. 4e,f). Overall, the LP-motif forms a PPII helix and binds to the RAC–SBD *in cis* like a pseudo-substrate.

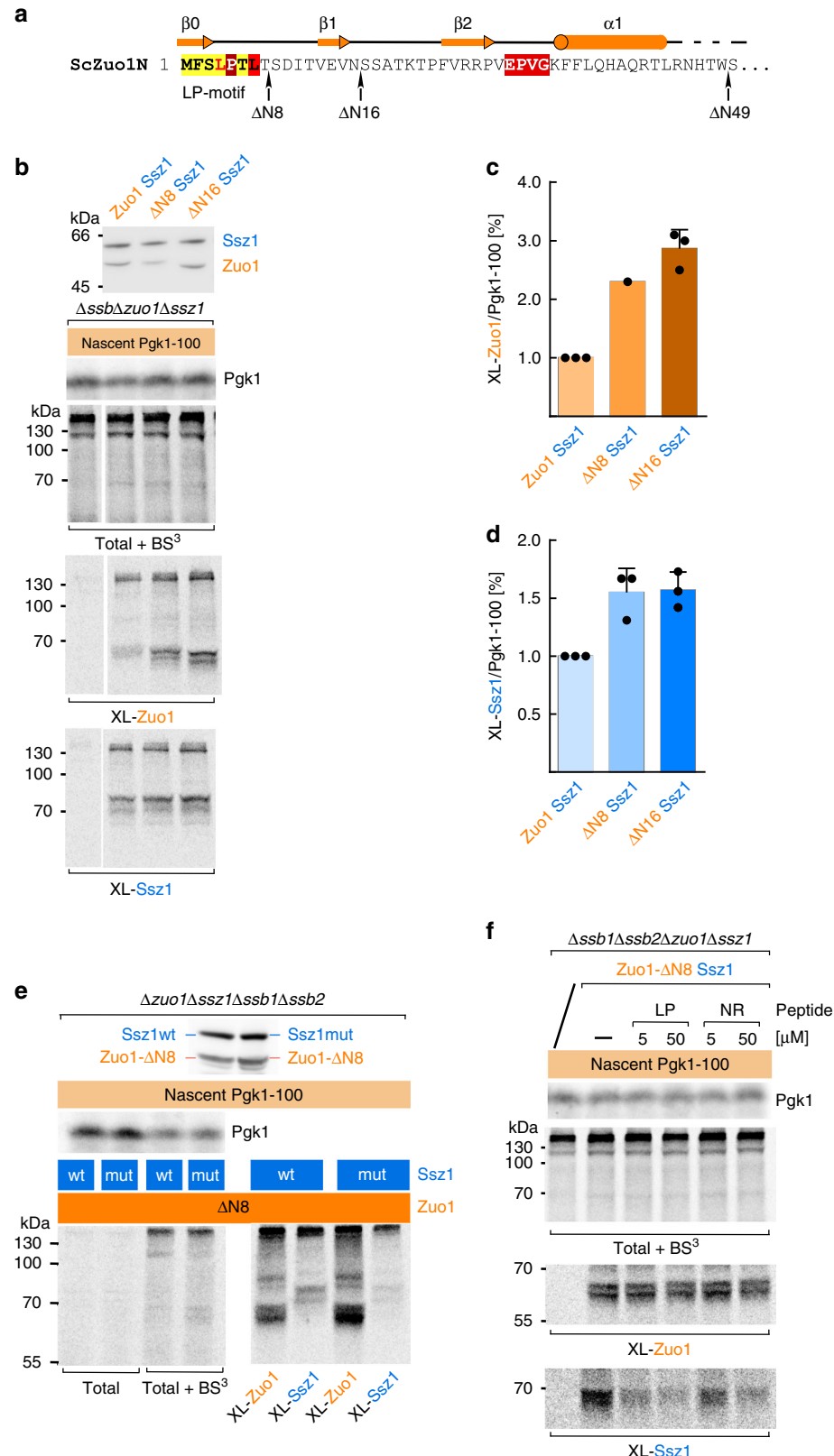

**The RAC–SBD prefers the LP-motif over the NR-peptide.** To analyze the RAC–SBD interaction with the LP-motif in more detail and to relate it to nascent chain binding in *Sc*RAC (see below), we designed two *Sc*RAC variants comprising *Sc*Ssz1 and either the *Sc*Zuo1N (residues 1-50, corresponding to *Ct* residues 1-60) or *Sc*Zuo1NΔLP (residues 10-50) lacking the LP-motif.

Please note that while the LP-motif is conserved from fungi to human, the Zuo1N β0 strand is shorter or absent in the human homolog MPP11 and in yeast (Fig. 2a). Nano differential scanning fluorimetry (nanoDSF) showed that deletion of the LP-motif drastically decreased the thermal stability of the RAC core complex from 52.9 °C to 39.7 °C (Supplementary Fig. 5a). Thus,

**Fig. 3 Nascent chain binding to the Ssz1-SBD is modulated by the N-terminus of Zuo1. a** Amino acid sequence of the *Sc*Zuo1N. Arrows indicate the positions of truncations in Zuo1-ΔN8, Zuo1-ΔN16, and Zuo1-ΔN49. The color code is as in Fig. 2a. The secondary structure derived from the crystal structure is depicted above. **b** The Zuo1N LP-motif interferes with the contact of nascent Pgk1-100 with Zuo1 and Ssz1. Ribosome-free extracts obtained from Δ*ssb1*Δ*ssb2* (Zuo1, Ssz1), Δ*ssb1*Δ*ssb2* RAC-ΔN8 (ΔN8, Ssz1), or Δ*ssb1*Δ*ssb2* RAC-ΔN16 (ΔN16, Ssz1) strains (upper panel) were added to RNCs from Δ*ssb1*Δ*ssb2*Δ*zuo1*Δ*ssz1* translation extract prior to crosslinking. Crosslink products between Zuo1 and nascent Pgk1-100 (XL-Zuo1) and Ssz1 and nascent Pgk1-100 (XL-Ssz1), respectively. **c, d** Quantification of crosslinking between nascent Pgk1-100 and Zuo1 (**c**) and Ssz1 (**d**). Autoradiographs as shown in (**b**) were employed for the quantification of nascent Pgk1-100 and the crosslink products XL-Zuo1 and XL-Ssz1. Values were normalized as described in Methods. The efficiency of crosslinking is given as a percentage of nascent Pgk1-100 present in the reaction. Experiments were performed in triplicate, with the exception of the XL-Zuo1-ΔN8 (shown in **c**), which was performed only once. Shown is the mean (bars) and the result of each biologically independent experiment (dots). Error bars indicate the standard deviation of the mean. **e** Crosslinking of the nascent chain to Ssz1 requires a functional peptide binding groove. Ribosome-free extracts from Δ*ssb1*Δ*ssb2* RAC-ΔN8 (Zuo1-ΔN8, Ssz1) or Δ*ssb1*Δ*ssb2* RACmut-ΔN8 (Zuo1-ΔN8, Ssz1mut) (upper panel) was added to RNCs generated in Δ*ssb1*Δ*ssb2*Δ*zuo1*Δ*ssz1* translation extract as described in Fig. 1b. Crosslink products between Zuo1 and nascent Pgk1-100 (XL-Zuo1) and Ssz1 and nascent Pgk1-100 (XL-Ssz1), respectively. **f** The LP-peptide displaces the nascent chain from Ssz1. RNCs were generated in a Δ*ssb1*Δ*ssb2*Δ*zuo1*Δ*ssz1* translation extract and ribosome-free extract from Δ*ssb1*Δ*ssb2* RAC-ΔN8 (Zuo1-ΔN8, Ssz1) was added. Subsequently, RNCs were isolated as described in Methods and LP-peptide (LP) or NR-peptide (NR) was added as indicated to the resuspended RNCs prior to the crosslinking reaction. Crosslink products between Zuo1 and nascent Pgk1-100 (XL-Zuo1) and Ssz1 and nascent Pgk1 (XL-Ssz1), respectively. Source data for (**b**–**f**) are provided as a Source Data file.

binding of the LP-motif to the RAC–SBD as a pseudo-substrate strongly stabilized the complex. We next asked whether free LP-peptide (MFSLPTL, corresponding to the LP-motif, *Sc*Zuo1 residues 1-7) was able to bind and thereby stabilize the complex. Indeed, addition of LP-peptide increased the melting temperature of Ssz1-Zuo1NΔLP from 40.3 to 42.9 °C (Supplementary Fig. 5b). This moderate, but significant effect on the thermal stability strongly suggested that the LP-peptide was able to bind to the Ssz1Zuo1NΔLP complex *in trans* (see also below).

In order to quantify the interaction, we employed fluorescence anisotropy and compared the LP-peptide and the NR-peptide with respect to binding to the Ssz1-Zuo1N and Ssz1-Zuo1NΔLP complex, respectively (Supplementary Fig. 6). While the affinity of the NR-peptide for both complexes was below the detection limit, the LP-peptide bound to the Ssz1-Zuo1NΔLP complex with an estimated affinity of 11.9 ± 4.0 μM, which is similar to the NR-peptide interacting with ATP-bound DnaK (23.8 ± 3.8 μM, low-affinity state[29]). In order to confirm that the LP-peptide was indeed interacting with the peptide binding groove of Ssz1, a mutant was generated in which four point mutations (L439S/K440P/I448F/G495K; Ssz1mut) were introduced into the peptide binding groove (Supplementary Fig. 5c). These mutations were based on previously characterized DnaK mutants deficient in substrate binding[30]. Analysis by nanoDSF showed that, in contrast to Ssz1-Zuo1NΔLP, the melting temperature of Ssz1mut-Zuo1NΔLP was basically not changed upon addition of the LP-peptide (Supplementary Fig. 5d). These data show that the LP-motif forming a PPII helix binds to the Ssz1-SBD as a low-affinity pseudo-substrate, and suggest that this motif is optimized in sequence, structure, and positioning for binding to the RAC–SBD *in cis*. We hypothesize that the strictly conserved Pro11 (Pro5 in *S. cerevisiae*) plays a central role in restricting the conformation of the LP-motif for optimal binding. Therefore, the RAC–SBD seems to prefer the LP-motif over the Hsp70 model substrate NR-peptide.

**The Zuo1 LP-motif modulates nascent chain binding to Ssz1.** As the LP-motif occupies the RAC–SBD *in cis* and competes with peptide binding *in trans* (Supplementary Fig. 6), we speculated that the LP-motif might compete also with nascent chain binding to ribosome-bound RAC. To test this directly, we generated yeast strains expressing Zuo1-ΔN8, which carried a deletion of the LP-motif, or Zuo1-ΔN16, which lacked the LP-motif and Zuo1N β1 (Fig. 3a). Zuo1-ΔN8 and Zuo1-ΔN16 formed stable complexes with Ssz1 (termed RAC-ΔN8 and RAC-ΔN16) (Supplementary Fig. 1e) and complemented growth defects of a Δ*zuo1* strain when

expressed in vivo (Supplementary Fig. 1i,j). The interaction of RAC-ΔN8 and RAC-ΔN16 with nascent Pgk1-100 was tested in the absence of Ssb, i.e. under conditions, in which wild type Zuo1 and Ssz1 efficiently interact with nascent chains (Fig. 1d and Supplementary Fig. 1b). Crosslinking of the nascent chain to both RAC subunits was enhanced in RAC-ΔN8 or RAC-ΔN16 complexes (Fig. 3b–d). To test if the nascent chain was indeed bound to the peptide binding groove of Ssz1 when Zuo1 lacked the LP motif, we compared the crosslinking efficiency of the nascent chain to RAC-ΔN8 (Zuo1-ΔN8/Ssz1) with the crosslinking efficiency to RACmut-ΔN8 (Zuo1-ΔN8/Ssz1mut) (Fig. 3e). Indeed, the contact between the nascent chain and Ssz1mut (Supplementary Fig. 5c,d) was strongly reduced when compared to Ssz1 (Fig. 3e). At the same time crosslinking of the nascent chain to Zuo1-ΔN8 was enhanced (Fig. 3e). These observations are consistent with a model that the nascent chain is transferred from Zuo1 to the Ssz1-SBD. We next tested if free LP- or NR-peptide, added before the crosslinking reaction, affected nascent chain interaction with the RAC subunits. Neither of the peptides added prior to the crosslinking reaction affected nascent chain crosslinking to Zuo1-ΔN8 (Fig. 3f, XL-Zuo1). However, addition of LP-peptide, and to a lesser extent also NR-peptide, significantly reduced crosslinking of the nascent chain to Ssz1 (Fig. 3f, XL-Ssz1). Thus, the LP-peptide was not only able to bind to the Ssz1-Zuo1NΔLP complex *in trans* (Supplementary Fig. 6), but also displaced a nascent chain from Ssz1 in the RAC-ΔN8 complex, indicating that both substrates competed for the same binding site (Fig. 3f). Based on the combined structural and biochemical data we conclude that enhanced proximity of the nascent chain to Zuo1-ΔN8/Zuo1-ΔN16 was likely due to structural and/or positioning effects. In contrast, enhanced proximity of the nascent chain to Ssz1 reflected *bona fide* nascent chain binding to Ssz1-SBDβ resembling the low-affinity interaction of ATP-bound canonical Hsp70s with their substrates.

## Discussion

In this work, we uncover mechanistically important features of the ribosome-bound RAC/Ssb chaperone system. Combining crosslinking experiments with high resolution structure determination and substrate binding assays, we answer the long-standing question whether the role of Ssz1 is to structurally support the function of the J-domain protein Zuo1, or whether it acts as a chaperone. We show that Ssz1 indeed has chaperone activity, that Ssz1-SBDβ interacts with nascent chains and that this interaction is modulated by a conserved motif in Zuo1N (LP-motif), which binds to the Ssz1-SBD like substrates bind to typical

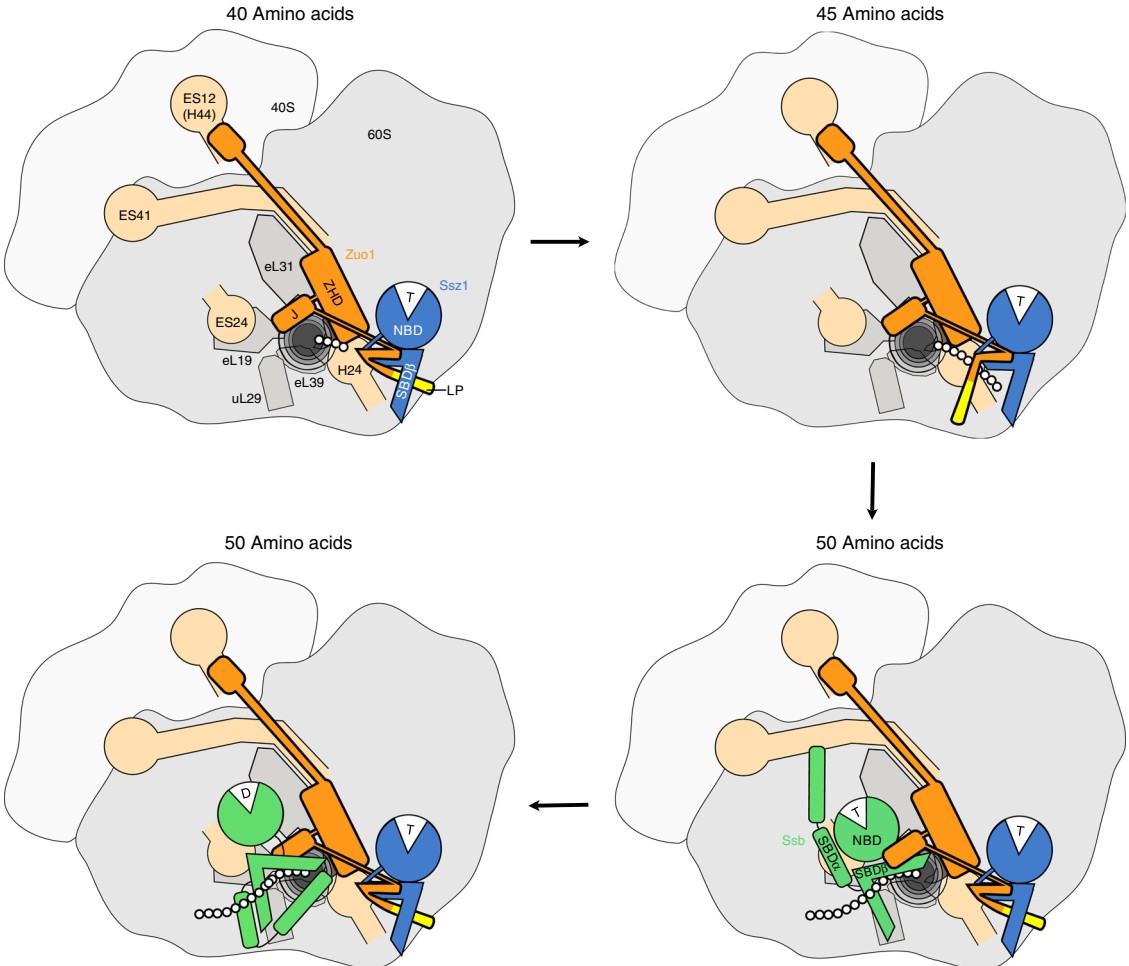

**Fig. 4 Nascent chain handover at the ribosomal tunnel exit.** Model of the RAC/Ssb substrate relay system at the ribosomal tunnel exit. At a length of 40 residues the nascent chain contacts Zuo1 (upper left panel). At a length of 45 residues the nascent chain reaches Ssz1 (upper right panel). This requires the displacement of the Zuo1 LP-motif from the Ssz1-SBDβ. At 50 residues length, the nascent chain is transferred to Ssb, in a process that requires a direct transfer from the Ssz1-SBDβ to Ssb-SBDβ, and a contact of the Zuo1 J-domain with the linker region of Ssb (lower right panel). The latter stimulates ATP hydrolysis by Ssb and traps the nascent chain in a Ssb·ADP·nascent chain complex (lower left panel). Likely in this situation the LP-motif interacts with the Ssz1-SBDβ. The ribosomal 60S and 40S subunits are shown in dark and light gray. The tunnel exit is indicated in black, and residues of nascent Pgk1 are represented as white circles. Zuo1 (orange), Ssz1 (blue), Ssb (green), and Zuo1 LP-motif (yellow). J-domain (J), Zuotin homology domain (ZHD), nucleotide binding domain (NBD), substrate binding domain (SBD). Main contacts of RAC and Ssb with ribosomal proteins (gray) and RNA (sand) are indicated (for details on the ribosomal contacts of RAC and Ssb see[4]). For more details compare Discussion.

Hsp70s. To demonstrate that the growing nascent chain is transferred from Ssz1 to Ssb, the use of nascent chains shorter than 50 residues (Fig. 4) was fundamental. Longer nascent chains were efficiently captured by Ssb and for this reason were no longer in contact with RAC.

Ssz1 is commonly regarded a non-canonical Hsp70, because it lacks the SBDα lid domain[4,17] and the Ssz1-NBD can bind, but cannot hydrolyze ATP[9,14]. The structure of the RAC core presented in this work reveals a novel property of Ssz1: Ssz1-SBDβ interacts with the Zuo1 LP-motif, which serves as a low-affinity inbuilt substrate. The consequences of these Ssz1-specific deviations from the general Hsp70 scheme can be understood by recalling that compared to wild type DnaK, a DnaKΔlid variant is destabilized and shows reduced substrate affinity and an 10–100-times faster $k_{on}$ and $k_{off}$[31,32]. Similarly, when the Hsp70 homolog BiP is AMPylated, ATP hydrolysis is inhibited and BiP is trapped in a low-substrate affinity ATP-like state[33]. In these cases, absence of the lid or inhibition of lid closure induces a destabilization of the SBD resulting in low-affinity substrate binding. Destabilization of Ssz1, induced by the loss of the lid domain, seems to be

compensated by Zuo1N in two ways: by additional β-strands of Zuo1 that embrace the Ssz1-SBD, and by positioning of the LP-motif within the substrate binding site of the Ssz1-SBD. These stabilizing effects, which Zuo1 imposes on Ssz1, at the same time contribute to the trapping of Ssz1 in the low-substrate affinity, ATP-bound state. Our findings reveal that Ssz1 displays chaperone activity, which is seemingly optimized for the highly transient interactions required to cope with the situation at the ribosomal tunnel exit, where nascent chain substrates emerge at a rate of about 3-6 residues per second[34,35]. The specific properties of Ssz1 allow for a rapid forward transfer to Ssb within the RAC/Ssb chaperone triad. In line with this notion the Zuo1 LP-motif negatively modulates nascent chain binding to Ssz1. This previously unknown function of Zuo1 in some respect resembles a recently discovered supplementary function of the yeast Hsp70 nucleotide exchange factor (NEF) Fes1[36]. Fes1 does not only act as a NEF, but via its flexible N-terminal domain mimics Hsp70 substrates. Accordingly, binding of the Fes1 N-terminal domain to the Hsp70-SBD promotes substrate release[36]. Ssz1 does not hydrolyze ATP and thus does not require a classical NEF

as a co-chaperone. The above mentioned supplementary function of NEFs[36], however, is seemingly adopted by the N-terminal LP-motif within the J-domain protein Zuo1. In RAC, the intrinsic LP-motif is positioned close to the Ssz1-SBD, suggesting that permanent competition between the LP-motif and the emerging nascent chain safeguards the release of substrates and further supports forward transfer to Ssb. Interestingly, the LP-motif forms a PPII helix and its binding mode is very similar to proline-rich antimicrobial peptides that target DnaK (Supplementary Fig. 4e,f)[26]. Notably, the PPII helix is an abundant and versatile recognition motif with a restricted backbone conformation, which allows for sequence specific recognition without requiring particularly high affinity interaction.

Based on the data presented and previously published work on the interaction of RAC and Ssb with the ribosome[4,10–12,22] we propose the following model for nascent chain handover at the ribosomal tunnel exit (Fig. 4).

At a length of 40 residues, an extended nascent chain predominantly contacts Zuo1. Because nascent Pgk1 becomes accessible at a length of 36 residues[21], at that point, only four residues are exposed and only the α-amino group of Met1 is available for crosslinking. (Fig. 4, upper left panel). Therefore, Zuo1 residue(s), which crosslink to Pgk1-40, should be within a radius of about 25 Å from the tunnel exit (considering that Met1 is about 15 Å from the exit and the BS[3] crosslinker bridges about 10 Å). However, as Zuo1 does not contain one of the conserved domains, which enable a sub-class of J-domain proteins to interact with substrate proteins[37], it is difficult to predict, which Zuo1 domain contacts the nascent chain. As Zuo1-Δ49 lacks the entire N-terminal region, but still crosslinks to the nascent chain, Zuo1N is not, or at least not the only, contact of Zuo1 with the nascent chain. One candidate is the Zuo1-ZHD, which directly contacts eL31 at the tunnel exit and is thus close to short nascent chains[12]. The ZHD is confined to close Zuo1 homologs and may thus potentially constitute a novel type of substrate binding domain. This possibility will be addressed in future studies.

At a length of 45 residues, Pgk1 was mostly in contact with Ssz1 (Fig. 4, upper right panel). This contact was observed only when Ssz1 was bound to Zuo1, indicating that Ssz1 had to be positioned at the tunnel exit to interact with a nascent chain. The unique properties of Ssz1 described in this study suggest that during translation, Ssz1-SBDβ does bind to the growing nascent chain, although not tightly, by that allowing for the continuous forward movement until the nascent chain gains contact with Ssb. As outlined above, Ssz1 seems optimized for transient low-affinity binding by specific deviations from the canonical Hsp70 mechanism. At a length of 50 residues, Pgk1 is mainly in contact with Ssb (Fig. 4, lower panels)[5–7]. Our data reveal that nascent Pgk1 was efficiently transferred from RAC to Ssb between a nascent chain length of 46–50 residues. Upon stimulation of the Ssb ATPase by the Zuo1 J-domain, Ssb switches to the high affinity conformation. If Ssb was absent, or was not bound to the ribosome, because it lacked its C-terminal ribosome-binding region (Ssb-ΔC23), or if Ssz1 was absent or its substrate binding site blocked by mutagenesis, the growing nascent chain remained bound to RAC or Zuo1, respectively. These observations suggested that the Ssz1-SBDβ has to be in close proximity to the Ssb-SBDβ to allow for efficient direct nascent chain transfer.

Seemingly the RAC/Ssb system of yeast is optimized for efficient nascent chain transfer at the ribosome. However, yeast cells cope well with conditions, in which direct nascent chain transfer at the ribosomal tunnel to Ssb is much reduced, e.g. when cells express Ssb-ΔC23[10,22]. A possible explanation for this, at first glance, puzzling observation is provided by the finding that RAC remains in contact with the growing nascent chain when Ssb is not ribosome-bound. One should consider that RAC is unlikely fixed permanently to a single ribosome, but cycles between ribosomes and the cytosol[4,5]. Thus, transfer of nascent chains from RAC to Ssb may not only occur directly at the tunnel exit, but also in the cytosol. Such transfer may, however, also directly go to Ssa. It is tempting to speculate that in mammalian cells a similar mechanism of de novo protein folding has evolved. Mammalian cells do not possess an Hsp70 homolog, which interacts with ribosomes directly[38]. However, mammalian cells possess a RAC homolog, termed mRAC, which, just like its yeast counterpart, is fully ribosome-associated in cell extract[39]. Nevertheless mRAC is the cochaperone of the cytosolic Hsp70 homolog HSPA1A/B[4,38,39]. Of note, when heterologously expressed in yeast, mRAC complements growth defects of Δzuo1Δssz1 cells. Complementation by mRAC is independent of Ssb, as mRAC, even though ribosome-associated, cooperates with the yeast cytosolic Hsp70 homolog Ssa[38]. Taken together our data provide a starting point for detailed analyses of nascent chain interaction with the RAC/Ssb triad and its homologs at the ribosome to finally derive mechanistic insights into their function in protein homeostasis.

## Methods

**Strains, plasmids, and growth conditions**. Strains and plasmids are listed in Supplementary Tables 1 and 2. All plasmids for expression of SSZ1 or ZUO1 variants in yeast contained 300 bp up- and down-stream of the respective *open reading frame*. The pRS315-Zuo1-ΔN8 was obtained by QuikChange Lightning site-directed mutagenesis (Agilent Technologies) employing pRS315-Zuo1[17] as a template. pRS423-Zuo1-ΔN8 was generated by transfer of Zuo1-ΔN8 from pRS315-Zuo1-ΔN8 to pRS423. The *CtZUO1* sequence encoding for residues 1–60 was PCR-amplified from pET16b-His$_6$-MBP-GSGSGS-TEV-CtZuo1[17] and was co-ligated with a PCR product of His$_6$-MBP with a 3C sequence overhang, which encodes a 3C (Human Rhinovirus 3C precision protease) cleavage site (LEVLFQ/GP), into a pET16b vector using the XbaI and BamHI sites resulting in pET16b-His$_6$-MBP-3C-CtZuo1N. The *Saccharomyces cerevisiae* sequence encoding for residues 1–50 of Zuo1 was amplified by PCR from pRS315-Zuo1[17] and was also co-ligated with the His$_6$-MBP-3C insert resulting in pET16b-His$_6$-MBP-3C-ScZuo1N. pET16b-His$_6$-MBP-3C-ScZuo1NΔLP (residues 10-50) was obtained by QuikChange Lightning site-directed mutagenesis (Agilent Technologies). The coding sequence of ScSSZ1 was amplified by PCR using Sc genomic DNA as a template and was cloned into pET24d-His$_6$-SUMO or pRS315, resulting in pET24d-His$_6$-SUMO-ScSsz1 or pRS315-ScSsz1. The L439S/K440P/I448F/G495K mutations were introduced into pYCPlac33-ScSsz1[19] by site-directed mutagenesis (QuikChange Lightning site-directed mutagenesis (Agilent Technologies)), resulting in pYCPlac33-Ssz1mut. ScSsz1 L439S/K440P/I448F/G495K was subsequently sub-cloned into pET24d-His$_6$-SUMO resulting in pET24d-His$_6$-SUMO-ScSsz1L439S/K440P/I448F/G495K (pET24d-His$_6$-SUMO-ScSsz1mut). Primers employed for cloning are shown in Supplementary Table 3.

The parental *Saccharomyces cerevisiae* wild type strain used in this study was MH272-3fα/a (ura3, leu2, his3, trp1, ade2)[40]. In the Δssz1 background, Zuo1 was expressed from a 2 μ plasmid to obtain a Zuo1 expression level similar to the wild type[8]. Yeast strains were grown in YPD complete medium (1% yeast extract, 2% peptone, 2% glucose), or in SD minimal medium (6.7 g/l yeast nitrogen base without amino acids, 2% glucose) supplemented with the appropriate amino acids and nucleobases. Cultures were grown in liquid medium at 30 °C with constant shaking at 200 rpm.

**Preparation of purified Zuo1 complexes**. pET16b-His$_6$-MBP-3C-CtZuo1N was coexpressed with pET24d-His$_6$-SUMO-strep-CtSsz1 and purified as described[17] with the following modifications. Dialysis was performed in 20 mM Hepes/KOH pH 7.5, 50 mM NaCl, 10 mM KCl, 5 mM MgCl$_2$ and 1 mM DTT containing 3C protease. The sample was then applied to a 1 ml ResourceQ column (GE Healthcare) and eluted with a 0–1 M NaCl gradient. The size-exclusion chromatography was performed in 20 mM Hepes/KOH pH 7.5, 150 mM NaCl, 10 mM KCl, 5 mM MgCl$_2$ and 1 mM DTT.

pET16b-His$_6$-MBP-3C-ScZuo1N or pET16b-His$_6$-MBP-3C-ScZuo1NΔLP was coexpressed with pET24d-His$_6$-SUMO-ScSsz1 or pET24d-His$_6$-SUMO-ScSsz1mut and purified as above with the following modified buffer composition. Dialysis buffer was 20 mM Hepes/KOH pH 7.5, 50 mM KOAc, 5 mM Mg(OAc)$_2$ and 1 mM DTT. The ResourceQ elution was performed with a 0–1 M KOAc gradient. The size-exclusion buffer was 20 mM Hepes/KOH pH 7.5, 120 mM KOAc, 5 mM Mg(OAc)$_2$ and 1 mM DTT.

ScSsb1 was expressed from pET24d-His$_6$-SUMO-Ssb1 and purified as above with the following modification. The nickel affinity chromatography column (IMAC) was washed with buffer containing 50 mM NaCl and 5 mM ATP and subsequently with buffer containing 1 M NaCl and 5 mM ATP. The eluate was

dialyzed against 20 mM Hepes/KOH pH 7.5, 120 mM KOAc, 5 mM Mg(OAc)$_2$ and 1 mM DTT in the presence of Ulp1 protease. The cleaved SUMO tag was removed by reverse IMAC, the Ssb1-containing flow-through was applied onto a ResourceQ and then eluted with a 0–1 M KOAc gradient. Ssb1-containing fractions were dialyzed against 20 mM Hepes/KOH pH 7.5, 120 mM KOAc, 5 mM Mg(OAc)$_2$ and 1 mM DTT, shock frozen in liquid nitrogen and stored at −80 °C.

**Crystallization and structure determination.** Crystallization screens were performed at 291 K by the sitting-drop vapor-diffusion method upon mixing equal volumes (0.2 µl) of the *Ct* Ssz1-Zuo1N protein solution (10 mg/ml) and reservoir solution containing 20.5% (v/v) PEG 3350 and 0.2 M ammonium acetate. Crystals grew after 21 h and were fished after 5 days. The crystals were cryo-protected by transfer into cryo-solution containing mother liquor and 20% (v/v) glycerol, and flash-frozen in liquid nitrogen. Diffraction data were measured under cryogenic conditions (100 K; Oxford Cryosystems Cryostream) at the European Synchrotron Radiation Facility (ESRF, Grenoble) beamline id30a3[41].

Data were processed with XDS[42]. Phases were obtained by molecular replacement using PHASER;[43] search model 5MB9[17] in the CCP4I2 software package[44,45]. Iterative model building and refinement were done with COOT[46] and phenix.refine[47]. The validation was performed using EDSTATS[48] and MOLPROBITY[49] included in CCP4I2. UCSF chimera[50] was used for figure preparation, always using chains C and D. Sequence alignments were performed using Clustal Omega[51] and visualized with ESPript 3.0;[52] http://espript.ibcp.fr/ESPript/ESPript/.

**Ribosome-binding assays.** Yeast strains were grown to early log phase on YPD, prior to harvest cycloheximide was added to a final concentration of 100 µg/ml, and subsequently cells were collected via centrifugation at 5000 *g*. Cell pellets were resuspended in ribosome-binding buffer (20 mM Hepes/KOH pH 7.4, 2 mM Mg (OAc)$_2$, 120 mM KOAc, 100 µg/ml cycloheximide, 2 mM DTT, 1 mM PMSF, protease inhibitor mix) and total cell extracts were prepared by the glass beads method[53]. After a clearing spin at 20.000 *g*, each 60 µl of the total glass beads extract (A$_{260}$ between 80 and 100 mAU) was loaded onto a 90 µl sucrose cushion (25% sucrose, 20 mM Hepes/KOH pH 7.4, 120 mM KOAc, 2 mM Mg(OAc)$_2$, 2 mM DTT, 1 mM PMSF, protease inhibitor mix). After centrifugation at 400.000 *g* at 4 °C for 25 min the cytosolic supernatant was collected and the ribosomal pellet was resuspended in 300 µl ribosome-binding buffer. Aliquots of the total cell extract, cytosolic supernatant, and resuspended ribosomal pellets were precipitated by addition of TCA (trichloroacetic acid) to a final concentration of 5%. TCA pellets were dissolved in SDS-sample buffer and were analyzed on 10% Tris-Tricine gels followed by immunoblotting.

**Preparation of yeast translation extract.** Yeast translation extract was prepared from 10 to 12 l cultures (OD$_{600}$ 0.8–1.0) of yeast strains as indicated in Results and Figure Legends. The protocol followed the method previously described by Walter and coworkers[54]. Collected cells were washed once with water and were then resuspended in 200 ml sorbitol buffer (1.4 M sorbitol, 50 mM K-phosphate buffer pH 7.4, 10 mM DTT) containing 2 mg zymolyase 20 T (nacalai tesque Inc.)/g cells. After incubation at 30 °C with agitation at 120 rpm for 30 min, spheroblasts were collected at 4 °C and were resuspended in 300 ml sorbitol buffer (YPD medium containing 1 M sorbitol). After incubation at 22 °C/120 rpm for 90 min, spheroblasts were re-collected, and were resuspended in 400 ml sorbitol buffer. The spheroblast suspension was transferred into two 500 ml JA-10 (Beckman Coulter) centrifuge tubes and was underlaid with 200 ml cold sorbitol buffer. After centrifugation at 4.000 *g* for 7 min at 4 °C in a JA-10 rotor, spheroblasts were washed twice with cold sorbitol buffer and were then resuspended in 5–10 ml lysis buffer (20 mM Hepes/KOH pH 7.4, 100 mM KOAc, 2 mM MgOAc$_2$, 2 mM DTT, 0.5 mM PMSF, 1× protease inhibitor mix). The spheroblast suspension was transferred into a 40 ml dounce homogenizer (Kontes Glass Co.) and spheroblasts were disrupted by douncing with a type B pestle on ice. The resulting extract was centrifuged in a SS34 rotor (Piramoon Technologies Inc.) at 15.000 rpm for 18 min at 4 °C. The supernatant was collected and was centrifuged in a 70.1 Ti rotor (Beckman Coulter) at 38.000 rpm for 35 min at 4 °C. The supernatant was loaded onto a Superdex-G25 gel filtration column equilibrated with gel filtration buffer (20 mM Hepes/KOH pH 7.4, 100 mM KOAc, 2 mM MgOAc$_2$, 2 mM DTT, 0.5 mM PMSF, 20% glycerol) at a flow rate of 1.5 ml/min. Peak fractions with the highest A$_{260}$ were pooled, were supplemented with 1 mM CaCl$_2$, and were treated with 300 U/ml micrococcal nuclease (Roche) for 15 min at 20 °C. Nuclease treatment was terminated by the addition of 2 mM EGTA and aliquots of the extract, termed yeast translation extract, were frozen in liquid nitrogen and were stored at −80 °C.

**In vitro transcription and translation.** DNA templates for transcription reactions were generated by PCR using pSPUTK-Pgk1[20] as a template. Reverse primers were designed such, that PCR products encoded different length *PGK1* sequences, which all lacked a stop codon. The length of the resulting translation products, which remain ribosome-bound, is indicated in Results and Figure Legends. Transcripts were generated using SP6 polymerase (Thermo Fischer Scientific) as previously described[55]. RNCs were generated via translation reactions primed with the stop codon-less transcripts at 20 °C in the presence of [$^{35}$S]-methionine

(Hartmann-Analytic) for 80 min[20,54]. Translation reactions were stopped by the addition of 50 µg/ml cycloheximide final concentration and subsequently RNCs were isolated via centrifugation at 400.000 *g* for 20 min in a TLA-100 or TLA-100.2 rotor (Beckman Coulter)[20].

**Chemical crosslinking and immunoprecipitation.** Chemical crosslinking of ribosome-bound nascent chains to Ssb, Zuo1, or Ssz1 was analyzed essentially as described[20]. In brief, resuspended RNCs were incubated in the presence of 400 µM BS$^3$ (*bis*-(sulfosuccinimidyl)-suberate, spacer length 1.14 nm, Thermo Scientific) for 20 min on ice and crosslinking was then quenched by the addition of TRIS base (tris(hydroxymethyl)aminomethane) to a final concentration of 50 mM. Cross-linked protein samples were precipitated by the addition of TCA (final concentration 5%), pellets were collected by centrifugation and were dissolved in dissociation buffer (200 mM TRIS/HCl pH 7.5, 4% SDS; 10 mM EDTA, 100 µg/ml BSA, protease inhibitor mix, 1 mM PMSF). Totals shown in Figs. 1,3 and Supplementary Fig. 1 before (total) and after BS$^3$ crosslinking (total + BS$^3$) represent 5% of the dissolved material employed for immunoprecipitation reactions. The resulting denatured protein samples were used to identify crosslink products between Ssb, Zuo1, or Ssz1 and nascent chains via affinity purification under denaturing conditions. To that end, protein A sepharose beads (GE Healthcare) were pre-coated with antibodies directed against Ssb, Zuo1, or Ssz1 and subsequently the denatured crosslinked material was allowed to bind to the beads resuspended in immunoprecipitation buffer (10 mM TRIS/HCl pH 7.5, 150 mM NaCl, 5 mM EDTA, 1% Triton X-100, protease inhibitor mix, 0.5 mM PMSF)[20]. Crosslink products bound to protein A sepharose beads were released by addition of SDS-sample buffer at 95 °C for 10 min. Samples were analyzed on Tris-Tricine gels[56] followed by autoradiography. Loading was normalized according to the signal of [$^{35}$S]-labelled nascent Pgk1 in the total prior to crosslinking. Low exposure autoradiographs of nascent chains in the totals (labelled Pgk1) are shown in Fig. 1 and Fig. 3. Relative crosslinking efficiencies between nascent Pgk1-100 and Zuo1 or Ssz1, respectively were determined as follows. For each biological replicate, band intensities of non-crosslinked nascent Pgk1-100 and of Pgk1-100 crosslink products (XL-Zuo1 or XL-Ssz1) was determined on a single autoradiograph for wild type RAC and the different mutant variants of RAC. The crosslinking efficiency (XL-Zuo1/Pgk1-100 or XL-Ssz1/Pgk1-100) of each sample was calculated and subsequently the ratio of XL-Zuo1/Pgk1-100 or XL-Ssz1/Pgk1-100 in the presence of wild type RAC was set to 1 in order to normalize the biological replicates (see Fig. 3c,d). The original data are provided as a Source Data file.

**Preparation of ribosome-free extracts.** For complementation experiments translation reactions were performed in either Δ*ssb1*Δ*ssb2* or Δ*ssb1*Δ*ssb2*Δ*zuo1*Δ*ssz1* translation extract as indicated. Complementation with purified Ssb1 was performed by adding His$_6$-tagged *Sc*Ssb1 at a final concentration of 2 µM to the translation reaction prior to the addition of mRNA. Ribosome-free cytosolic extract of various mutant yeast strains for complementation experiments as indicated in Results and Figure Legends was generated as follows. The A$_{260}$ of a total cell extract prepared by the glass beads method (see above) was determined, and the total cell extract was subsequently adjusted to a final concentration of 800 mM KOAc. Under this condition ribosome-bound proteins such as Ssb, Zuo1 and Ssz1 are released from ribosomes. Ribosomes were then collected by ultracentrifugation at 400.000 *g* for 25 min at 4 °C and the resulting supernatant (ribosome-free extract) was re-adjusted to a concentration of 120 mM KOAc by addition of KOAc-free resuspension buffer (20 mM Hepes/KOH pH 7.4, 2 mM Mg(OAc)$_2$, 50 µg/ml cycloheximide, 1 mM PMSF, protease inhibitor mix). Ribosome-free extract was added to Ssb/RAC-free RNCs after completion of translation reactions. The amount of ribosome-free extract added to RNCs was adjusted such that the total A$_{260}$ of the added ribosome-free extract equaled the A$_{260}$ of the yeast translation reaction. After incubation for 5 min at 20 °C, RNCs and bound factors were isolated via sedimentation at 400.000 *g* and were resuspended in 300 µl resuspension buffer prior to crosslinking analysis, which was performed as described above.

**Binding affinity measurements using fluorescence anisotropy.** The protein buffers were exchanged for anisotropy buffer (20 mM Hepes/KOH pH 7.5, 200 mM KCl, 5 mM MgCl$_2$, 1 mM DTT) using Zeba spin columns (Thermo Scientific). 1 µM BSA was added to prevent unspecific binding of the fluorescently labelled peptide to the microplate (Greiner Bio-One opaque 384 well plate). Fluorescence anisotropy measurements were performed using the LP-peptide (MFSLPTL) or the NR-peptide (NRLLLTG) labelled at the N terminus with fluorescein (Peptide Specialty Laboratories GmbH, Heidelberg). Two-fold serial dilutions of each protein (*Sc*Ssz1-Zuo1N or *Sc*Ssz1-Zuo1NΔLP) were mixed in a 1:1 ratio with 40 nM peptides in anisotropy buffer in 384-well opaque plates (Greiner Bio One) and were incubated at room temperature for 30 min. Measurements were performed in triplicate using a plate reader (SpectraMax M5e Multi Mode Microplate reader, Molecular Devices). To determine the binding constant ($K_D$) the data were fitted to a one-site binding equation using Python[57].

**NanoDSF measurements.** The melting temperature of *Sc*Ssz1-Zuo1N, *Sc*Ssz1-Zuo1NΔLP or *Sc*Ssz1mut-Zuo1NΔLP with or without the LP-peptide (Peptide Specialty Laboratories GmbH, Heidelberg) were determined using the Prometheus

NT.48 nanoDSF system using nano-DSF grade capillaries (NanoTemper Technologies GmbH). The measurements were performed in 20 mM Hepes/KOH pH 7.5, 200 mM KCl, 5 mM MgCl$_2$, 1 mM DTT. The LP-peptide, which was dissolved in DMSO, was pre-diluted in the above buffer such that the final DMSO concentration in the experiment was below 0.5%. DMSO diluted in the same way was added to the negative control. Intrinsic protein fluorescence was measured continuously at 330 nm and 350 nm with a temperature gradient ranging from 20 to 90 °C at a rate of 1.5 °C/min.

**Statistics and Reproducibility**. Figures 1, 3, and Supplementary Fig. 1 include representative autoradiographs and immunoblots. The experiments were performed at least for the following number of times: Fig. 1c, two times; Fig. 1d, two times; Fig. 1e, three times; Fig. 1f, three times; Fig. 3b–d, see Figure Legend and Source Data file; Fig. 3e, two times; Fig. 3f two times; Supplementary Fig. 1a, two times; Supplementary Fig. 1b, two times; Supplementary Fig. 1c, three times; Supplementary Fig. 1d, one time; Supplementary Fig. 1e, two times; Supplementary Fig. 1f, two times; Supplementary Fig. 1g, two times; Supplementary Fig. 1h, three times; Supplementary Fig. 1i, two times; Supplementary Fig. 1j, two times; Supplementary Fig. 6, three times.

**Miscellaneous**. Proteins were separated on 10% Tris-Tricine gels[56]. Antibodies for immunoblotting and denaturing immunoprecipitation reactions are rabbit polyclonal antibodies (Rospert lab antibody collection). Antibody dilutions were as follows: α-Ssb 1:5.000, α-Ssz1 1:20.000, α-Zuo1 1:40.000, α-Sse1 1:20.000, α-Rps9 1:8.000, α-Rpl4 1:20.000, and α-Pgk1 1:20.000. Horseradish-conjugated goat anti-rabbit IgG (Pierce catalog number 31460, 1:10.000) was employed as a secondary antibody. Immunoblots were developed by enhanced chemiluminescence[58]. Protease inhibitor mix (1×) contained 1.25 μg/ml leupeptin, 0.75 μg/ml antipain, 0.25 μg/ml chymostatin, 0.25 μg/ml elastinal, and 5 μg/ml pepstatin A. Quantification of digital phosphorimages (autoradiographs) was performed using AIDA ImageAnalyzer (Raytest). At least 3 independent biological replicates were performed for statistical analysis, which was conducted with GraphPad Prism (version 6.07).

**Reporting summary**. Further information on experimental design is available in the Nature Research Reporting Summary linked to this paper.

## Data availability

Accession codes: Coordinates and structure factors have been deposited in the Protein Data Bank under accession code 6SR6. The PDB datasets 5MB9, 1DKZ and 4EZP have been used in this study. The source data underlying Figs. 1c-f, 3b-f, and Supplementary Figs. 1a-j, 5a-c, and 6a,b are provided as a Source Data file. Other data that support the findings of this study are available from the corresponding authors on reasonable request.

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

## Acknowledgements

The authors thank C. Siegmann from the BZH/Cluster of Excellence:CellNetworks crystallization platform for protein crystallization, and acknowledge access to the beamlines at the European Synchrotron Radiation Facility (ESRF) in Grenoble and the support of the beamline scientists. We acknowledge A. Hendricks, J.P. Kreysing, M. McDowell and D. Layer for technical assistance. We thank G. Stier for providing plasmids and K. Wild for valuable comments on the structure and the manuscript. This work was funded by the Deutsche Forschungsgemeinschaft (DFG) through the Leibniz programme (SI 586/6-1) and Project-ID 201348542 - SFB 1036, TP22 (to I.S.), Project-ID 403222702 - SFB 1381, TP B08 (to S.R.), and RO 1028/5-1 (to S.R.), and by the Excellence Initiative of the German federal and state governments [BIOSS-2] (to S.R) and [CellNetworks] (to I.S.). I.S. is an investigator of the Cluster of Excellence: CellNetworks.

## Author contributions

Y.Z., G.V.G., C.C., K.L., S.R., and I.S. designed the experiments and analyzed the data. G. V.G. and J.K. solved the crystal structure. Y.Z., G.V.G., C.C., T.W., and K.L. performed experiments. Y.Z., G.V.G., K.L, S.R., and I.S. wrote the manuscript. All authors commented on the manuscript.

## Competing interests

The authors declare no competing interests.
