## [Peer Review File · Nature Communications]

Reviewers' comments:

Reviewer #1 (Remarks to the Author):

"The ribosome-associated complex RAC serves in a relay that directs nascent chains to Ssb"

Zhang et al. provide new insights into the interaction of Ssz1 and Zuo1 within the RAC complex and the role of this interaction in RACs function on the ribosome. Based on their biochemical and structural data they suggest a hand-over mechanism for nascent chains at the ribosomal tunnel exit from Zuo1 via Ssz1 to Ssb1. In contrast to previous studies, they show via crosslinking assays that not only Zuo1 but also Ssz1 contacts short nascent chains (40 or 45 aa long) on the ribosome before the latter start to contact the ribosome associated chaperone Ssb (starting at 50 aa length). In the absence of Ssb both RAC subunits also contact longer nascent chains. The authors show that both RAC subunits are required for the handover of the nascent chain substrate from RAC to Ssb.

Zhang et al. identify a previously unrecognized conserved motif (LP motif) in the N-terminus of Zuo1 and show via x-ray crystallography that this N-terminus of Zuo1 contributes two additional β -strands to the substrate binding domain (SBD β) of Ssz1, and, more over that the LP motif binds as a pseudo-substrate within the SBD of Ssz1 leading to a stabilization of the complex as shown by nanoDSF. The structure reveals that the LP motif forms a polyproline-II helix optimally adjusted for low affinity binding to the SBD binding site. The data show that Ssz1 does not only structurally support the function of the J-domain protein Zuo1, but actively participates as a chaperone in the folding of newly emerging nascent chains on the ribosome.

The study provides new insights into the function of the chaperone triad composed of Zuo1, Ssz1 and Ssb on the ribosome. The biochemical and structural data are very clear and strongly support the proposed new model that RAC serves as a substrate relay system on the ribosome.

In summary, the manuscript provides a level of novelty and common interest, which fully justifies the publication in Nature Communications.

I therefore suggest publication if some minor points are addressed:

Page 2 line 62

"RAC core (consisting of Ssz1 and Zuo1N19-60) shows..."

Should rather be "RAC core (consisting of Ssz1 and Zuo1N residue 19 to 60) shows..."

Page 3 line 87

It would be helpful for the reader if the cross-linker and its spacer length would be mentioned here.

Page 3 line 90

"Consistent with previous data 8, crosslinks between Zuo1 or Ssz1 and nascent Pgk1-50, or longer, were below the detection limit."

This sentence is a bit misleading since in reference 8 different nascent chains than Pgk1 were used.

Page 5 line 215

There is a "to" missing at the end of the line.

Fig. 3 b

Why is the expression level of Zuo1 Δ N8 so low ? Is this a stable protein ?

Fig. 3 e

In order to displace the nascent chain from Ssz1 very high concentrations (5 or 50 μ M) of LP or NR peptide were used, whereas the input of RNCs seems to be much lower than in the other experiments. In order to show that the observed displacement of the nascent chain is not simply

caused by the massive excess of peptide, a control should be provided where similar concentrations of an unrelated (non-substrate) peptide are present in the assay.

Reviewer #2 (Remarks to the Author):

Ribosome-associated molecular chaperones contribute to the biogenesis of proteins in all domains of life. In eukaryotic cells, the ribosome-associated complex (RAC) binds to the ribosomal exit tunnel. In yeast, it consists of the J-domain protein Zuo1 and the non-canonical Hsp70 paralogue Ssz1. Metazoans have distantly related homologues to RAC. RAC collaborates with the ribosome-binding Hsp70 orthologue Ssb in the stabilization of nascent chains and hand-over to downstream molecular chaperones in yeast. The J-domain in RAC stimulates ATP hydrolysis by Ssb, resulting in stable substrate interactions with Ssb. The three-dimensional structure of RAC and its interaction with the ribosome are fairly well known. Zuo1 recruits Ssz1 to the ribosome via interactions of its N-terminal domain, which completes the beta-sandwich domain of Ssz1. Ssz1 has no intrinsic ATPase activity and is missing the alpha-helical lid domain of canonical Hsp70 proteins. No clear function of Ssz1 in translation apart from scaffolding the Zuo1 J-domain has been found. Ssz1 might have a molecular chaperone / holdase function employing its beta-sandwich domain in analogy to canonical Hsp70 proteins, but proof for this hypothesis is missing.

Here the authors show that Zuo1 and Ssz1 crosslink to nascent Pdk1 chains of 40 and 45 residue length, whereas Ssb only crosslinks with nascent chains of 50 residues or longer, but only when RAC is present. Crosslinks to RAC are suppressed with these longer chains. The SDS-PAGE gels showing the crosslinking products furthermore suggest that Ssz1 preferentially interacts with the longer nascent chain. In summary, the data are consistent with a role of RAC in the hand-over of the nascent chain to Ssb.

Next, the authors present the crystal structure of the complex of Ssz1 with Zuo1(1-60) from *Chaetomium thermophilum*, which additionally includes Zuo1(1-18) compared to a previous structure. In the crystal lattice, residues 8-13 bind as a pseudo-substrate to the canonical Hsp70-substrate binding site in the beta-sheet domain of Ssz1. The affinity of the derived LP-peptide to yeast RAC is fairly low, but similar to substrate peptide interactions with canonical Hsp70 bound to ATP. The crosslinking properties of the delta(1-8) mutant are consistent with a regulatory role of the Zuo1 N-terminal peptide in displacing substrate from Ssz1. This resembles the function of the N-terminal domain in the Hsp70-nucleotide exchange factor Fes1 in yeast (Gowda et al., NSMB 2018). Direct experimental proof that the interaction of the nascent chain with the Ssz1 beta-sandwich domain as observed in the structure contributes to the hand-over to Ssb is missing.

While this study is exciting for all interested in protein translation and folding, I find the scope of the study quite narrow (the function of a seemingly non-essential segment of Zuo1) and the data too preliminary for publication at the moment.

The study would be more conclusive if the properties of Ssz1 mutants were studied in which the putative substrate binding site has been blocked by mutation. This might definitely prove that Ssz1 works as a molecular chaperone in the hand-over of substrates to Ssb. These mutants could be tested *in vitro* with the crosslinking assay and *in vivo* by employing the growth defect of the *ssz1*-deletion strain as background.

One could also test whether fusing the *Chaetomium*-Zuo1(1-18) to yeast Zuo1 delta(1-8), which might fortify the blockade of the putative binding site in Ssz1, impairs hand-over of nascent chains to Ssb in yeast.

Reviewer #3 (Remarks to the Author):

This is an interesting work uncovering the chaperone function of the unusual ribosome bound member of the yeast Hsp70 family, *ssz1* that was previously thought to be acting only as a cochaperone, and studying the dynamics and structural parameters of the nascent polypeptide chain binding by the members of the yeast ribosome associated chaperone and cochaperone triad, RAC-Ssb. The paper is nicely written, experiments are of high quality, and I'd expect this work to make a significant contribution to the field and beyond.

I do however have one serious issue that needs to be addressed at the revision stage. I believe authors should pay more attention to whether or not Ssb binding to the ribosome (both direct and indirect) is actually affected in their experiments, and controls for the Ssb association with the ribosome (rather than only to the nascent polypeptide) should be present. Indeed, for the experiments described on Lines 110-113 and Fig. 1d, authors kind of imply that Ssb binds to the ribosome only through its C-terminus. However, how do authors results fit with the previous observations that Ssb lacking the ribosome binding C-terminal region still binds to the ribosomes (presumably through Zuo1 and Ssz1) and participates in the nascent polypeptide folding? If authors suggest that this binding occurs at a different location, this should be addressed more thoroughly.

Even more importantly, for the experiments described in lines 121-126 and shown on Fig. 1, is Ssb indeed bound to the ribosome in these cells? As RAC regulates Ssb binding to the ribosome, it is possible that nascent chains are not transferred from Zuo1 to Ssb simply because because Ssb is no longer on the ribosome, rather than because of the absence of Ssz1 contact with Zuo1 per se. It is know that RAC disruption causes a release of *ssb* from the ribosomes to cytosol. In such a scenario, interaction between Zuo1 and Ssz1 would of course still play an important role, but in promoting Ssb association with the ribosome, rather than in transferring the nascent polypeptide chain to Ssb (that could be just a consequence of the Ssb/ribosome association). While not invalidating the overall importance of authors' data, such an explanation would significantly change the interpretation, thus I believe it has to be either addressed as an alternative explanation or ruled out by the additional experiments.

Point-by point answer to reviewers' comments:

Reviewer #1 (Remarks to the Author):

“The ribosome-associated complex RAC serves in a relay that directs nascent chains to Ssb”

Zhang et al. provide new insights into the interaction of Ssz1 and Zuo1 within the RAC complex and the role of this interaction in RACs function on the ribosome. Based on their biochemical and structural data they suggest a hand-over mechanism for nascent chains at the ribosomal tunnel exit from Zuo1 via Ssz1 to Ssb1. In contrast to previous studies, they show via crosslinking assays that not only Zuo1 but also Ssz1 contacts short nascent chains (40 or 45 aa long) on the ribosome before the latter start to contact the ribosome associated chaperone Ssb (starting at 50 aa length). In the absence of Ssb both RAC subunits also contact longer nascent chains. The authors show that both RAC subunits are required for the handover of the nascent chain substrate from RAC to Ssb.

Zhang et al. identify a previously unrecognized conserved motif (LP motif) in the N-terminus of Zuo1 and show via x-ray crystallography that this N-terminus of Zuo1 contributes two additional β -strands to the substrate binding domain (SBD β) of Ssz1, and, more over that the LP motif binds as a pseudo-substrate within the SBD of Ssz1 leading to a stabilization of the complex as shown by nanoDSF. The structure reveals that the LP motif forms a polyproline-II helix optimally adjusted for low affinity binding to the SBD binding site. The data show that Ssz1 does not only structurally support the function of the J-domain protein Zuo1, but actively participates as a chaperone in the folding of newly emerging nascent chains on the ribosome.

The study provides new insights into the function of the chaperone triad composed of Zuo1, Ssz1 and Ssb on the ribosome. The biochemical and structural data are very clear and strongly support the proposed new model that RAC serves as a substrate relay system on the ribosome.

In summary, the manuscript provides a level of novelty and common interest, which fully justifies the publication in Nature Communications.

We thank the referee for his/her positive comments and finding our data new and interesting.

I therefore suggest publication if some minor points are addressed:

Page 2 line 62

> “RAC core (consisting of Ssz1 and Zuo1N19--60) shows....”

> Should rather be “RAC core (consisting of Ssz1 and Zuo1N residue 19 to 60) shows....”

Done

> Page 3 line 87

> It would be helpful for the reader if the cross-linker and its spacer length would be mentioned here.

Thank you for this very good suggestion. We have now included the spacer length of BS³ in the Legend of Fig. 1b (Experimental set up of crosslinking experiments), which is introduced to the reader in line 81 (revised version).

> Page 3 line 90

> “Consistent with previous data 8, crosslinks between Zuo1 or Ssz1 and nascent Pgk1--50, or longer, were below the detection limit.”

> This sentence is a bit misleading since in reference 8 different nascent chains than Pgk1 were used.

We have made this paragraph more clear:

"Consistent with previous crosslinking analysis of nascent prepro α -factor⁸, crosslinks between Zuo1 or Ssz1 and nascent Pgk1-50, or longer, were below the detection limit (Supplementary Fig. 1a)" (lines 84-87).

> Page 5 line 215

> There is a "to" missing at the end of the line.

Done (line 207)

Fig. 3 b

Why is the expression level of Zuo1 Δ N8 so low ? Is this a stable protein ?

The intensity of the Zuo1 Δ N8 band (and also of Ssz1) specifically on the immunoblot shown in **Fig. 3b** (upper panel) is indeed slightly lower when compared to the wild type. However, this was not a general observation and we have no indication for a destabilization of Zuo1 Δ N8. To that end, please compare the Zuo1 blots shown in **Supplementary Fig. 1c** and also **Supplementary Fig. 1g**. Please also note, that the level of Zuo1 Δ N16, which carries an even longer N-terminal deletion, is not reduced (**Fig. 3b upper panel**). Importantly, with respect to the experiment shown in Fig. 3b: even though possibly slightly less Zuo1 Δ N8 was added to RNCs in this experiment the XL- Δ N8Zuo1 was enhanced when compared to the XL-Zuo1. This effect would be further enhanced if more Zuo1 Δ N8 was added to the reaction.

Fig. 3 e

> In order to displace the nascent chain from Ssz1 very high concentrations (5 or 50 μ M) of LP or NR peptide were used, whereas the input of RNCs seems to be much lower than in the other experiments. In order to show that the observed displacement of the nascent chain is not simply caused by the massive excess of peptide, a control should be provided were similar concentrations of an unrelated (non-substrate) peptide are present in the assay.

Thank you for addressing these important issues.

1. The amount of RNCs in all experiments is similar.

The amount of RNCs in all experiments shown in this manuscript was very similar. The original phosphorimages were saved with parameters automatically adjusted by ImageQuant (GE Healthcare). In most cases we linearly adjusted the contrast for a more clear illustration. We had simply forgotten to adjust the contrast of nascent Pgk1-100 in the totals of this specific autoradiograph (**Fig. 3f**). This was now changed.

2. The peptide concentration applied in competition experiments is in the range of the Kd.

Our results indicate that the Zuo1 Δ N Δ LP/Ssz1 core complex displayed low affinity ($11.9 \pm 4.0 \mu$ M) for the LP-peptide. The affinity of the Zuo1 Δ N Δ LP/Ssz1 core complex for the NR peptide was below the detection limit. (see Results lines 212/213 and Supplementary Fig. 6). In order to detect displacement of the nascent chain the concentration of the peptide should be at least in the range of the Kd. We employed 5 μ M (~ 0.5-times Kd) and 50 μ M (~ 5-times Kd). One also needs to take into account that the nascent chain is "linked" to the ribosome in close proximity of Zuo1/Ssz1 and thus the local nascent chain concentration close to RAC was high, independent of the concentration of RNCs. In contrast, peptides (LP- and NR-peptide in our assay) were distributed equally in the reaction mix.

3. The NR-peptide serves as a "non-binding" control.

We have employed the NR-peptide as such a control, because we found that *in vitro* the affinity of the NR-peptide to the Zuo1N Δ LP/Ssz1 core complex was too low to be determined via fluorescence anisotropy measurements (see Results lines 212/213 and Supplementary Fig. 6). We found that when the Zuo1N Δ LP/Ssz1 was bound to a ribosome-bound nascent chain 50 μ M concentration of the NR-peptide partly displaced the nascent chain. Of note, displacement by the NR-peptide was significantly lower than displacement by the LP-peptide. This indicates that the Ssz1 subunit of RAC displays a low affinity for various peptides/nascent chain segments.

Reviewer #2 (Remarks to the Author):

Ribosome-associated molecular chaperones contribute to the biogenesis of proteins in all domains of life. In eukaryotic cells, the ribosome-associated complex (RAC) binds to the ribosomal exit tunnel. In yeast, it consists of the J-domain protein Zuo1 and the non-canonical Hsp70 paralogue Ssz1. Metazoans have distantly related homologues to RAC. RAC collaborates with the ribosome-binding Hsp70 orthologue Ssb in the stabilization of nascent chains and hand-over to downstream molecular chaperones in yeast. The J-domain in RAC stimulates ATP hydrolysis by Ssb, resulting in stable substrate interactions with Ssb. The three-dimensional structure of RAC and its interaction with the ribosome are fairly well known. Zuo1 recruits Ssz1 to the ribosome via interactions of its N-terminal domain, which completes the beta-sandwich domain of Ssz1. Ssz1 has no intrinsic ATPase activity and is missing the alpha-helical lid domain of canonical Hsp70 proteins. No clear function of Ssz1 in translation apart from scaffolding the Zuo1 J-domain has been found. Ssz1 might have a molecular chaperone / holdase function employing its beta-sandwich domain in analogy to canonical Hsp70 proteins, but proof for this hypothesis is missing.

Here the authors show that Zuo1 and Ssz1 crosslink to nascent Pgk1 chains of 40 and 45 residue length, whereas Ssb only crosslinks with nascent chains of 50 residues or longer, but only when RAC is present. Crosslinks to RAC are suppressed with these longer chains. The SDS-PAGE gels showing the crosslinking products furthermore suggest that Ssz1 preferentially interacts with the longer nascent chain. In summary, the data are consistent with a role of RAC in the hand-over of the nascent chain to Ssb.

Next, the authors present the crystal structure of the complex of Ssz1 with Zuo1(1-60) from *Chaetomium thermophilum*, which additionally includes Zuo1(1-18) compared to a previous structure. In the crystal lattice, residues 8-13 bind as a pseudo-substrate to the canonical Hsp70-substrate binding site in the beta-sheet domain of Ssz1. The affinity of the derived LP-peptide to yeast RAC is fairly low, but similar to substrate peptide interactions with canonical Hsp70 bound to ATP. The crosslinking properties of the delta(1-8) mutant are consistent with a regulatory role of the Zuo1 N-terminal peptide in displacing substrate from Ssz1. This resembles the function of the N-terminal domain in the Hsp70-nucleotide exchange factor Fes1 in yeast (Gowda et al., NSMB 2018).

Thank you for pointing out the interesting work of the Andréasson and Mayer groups in this context. Gowda et al. found that Fes1 does not only act as a nucleotide exchange factor (NEF) for Hsp70 but in addition also triggers substrate release from Hsp70. The work suggests that the N-terminal domain of Fes1 contacts the open SBD cavity of Hsp70 – just like the LP-motif of Zuo1 contacts the SBD of Ssz1 – and by that blocks rebinding of the substrate. In our case, the LP motif hinders substrate binding to the SBD and therefore ensures successful forward transfer to Ssb. Along these lines, we have added two sentences in the discussion (lines 290 to 300):

“In line with this notion the Zuo1 LP-motif negatively modulates nascent chain binding to Ssz1. This previously unknown function of Zuo1 in some respect resembles a recently discovered supplementary function of the yeast Hsp70 nucleotide exchange factor (NEF) Fes1³⁶. Fes1 does not only act as an NEF, but via its flexible N-terminal domain mimics Hsp70 substrates. Accordingly, binding of the Fes1 N-terminal domain to the Hsp70-SBD promotes substrate release³⁶. Ssz1 does not hydrolyze ATP and thus does not require a classical NEF as a co-chaperone. The above mentioned supplementary function of NEFs³⁶, however, is seemingly adopted by the N-terminal LP-motif within the J-domain protein Zuo1. In RAC, the intrinsic LP-motif is positioned close to the Ssz1-SBD, suggesting that permanent competition between the LP-motif and the emerging nascent chain safeguards the release of substrates and further supports forward transfer to Ssb. “

Direct experimental proof that the interaction of the nascent chain with the Ssz1 beta-sandwich domain as observed in the structure contributes to the hand-over to Ssb is missing.

Very important points raised by the referee. However, the displacement of the nascent chain by the LP-peptide as indicated by the loss of crosslink is strong evidence that both ligands compete for binding to the SBD. If this would not be the case, the LP peptide should not have an effect on the XL. In the Fes1 paper they seem to do in fact a similar experiment as the N-terminal domain (RD) blocks NR peptide binding to Ssa1 and when RD is deleted there is no effect on NR binding.

Along these lines, we performed additional experiments as suggested by this referee (see below) to further substantiate substrate binding to the SBD. Based on information from substrate binding deficient DnaK mutants (Montgomery et al. 1999) residues within the substrate binding groove of Ssz1 were replaced (L439S-K440P-I448F-G495K) resulting in Ssz1mut (**Supplementary Fig. 5c**). Analysis via nanoDSF revealed that Zuo1NΔLP/Ssz1mut was no longer stabilized by LP-peptide, indicating that the LP-peptide was unable to bind to Ssz1mut (new **Supplementary Fig. 5d**). In addition, nascent chain crosslinking to Ssz1mut was impaired when compared to nascent chain crosslinking to wild type Ssz1. The data reveal that the nascent chain was indeed crosslinked to the substrate binding groove of Ssz1 (**Fig. 3e**). We included these new data as **Fig. 3e** and in lines 214 to 220 and lines 239 to 246, respectively. We feel that the combined data, now including the new data shown in **Fig. 3e**, provide strong and direct evidence for a hand-over mechanism that involves the Ssz1-SBD.

While this study is exciting for all interested in protein translation and folding, I find the scope of the study quite narrow (the function of a seemingly non-essential segment of Zuo1) and the data too preliminary for publication at the moment.

Here, we respectfully disagree with the referee. The scope of the study is not the function of the N-terminal segment of Zuo1, although we find it important to reveal a function for this poorly characterized domain, which is confined to close homologs of Zuo1.

Main scope of the study is:

- to show (for the first time) that Ssz1 interacts with nascent chains.
- to show that Ssz1 is actually optimized for the efficient and continuous transfer of nascent chains to Ssb.
- provide structural data which
 - show that the “minimal” SBD of Ssz1 does bind a substrate just like canonical Hsp70s do.
 - unravel the role of the Zuo1 N-terminal domain as a pseudo substrate which stabilizes the Ssz1-SBD and modulates substrate transfer.

We find these new insights of high general importance as they deepen our understanding of the RAC chaperone system.

The study would be more conclusive if the properties of Ssz1 mutants were studied in which the putative substrate binding site has been blocked by mutation. This might definitely prove that Ssz1 works as a molecular chaperone in the hand-over of substrates to Ssb. These mutants could be tested *in vitro* with the crosslinking assay and *in vivo* by employing the growth defect of the *ssz1*-deletion strain as background. One could also test whether fusing the Chaetomium-Zuo1(1-18) to yeast Zuo1 delta(1-8), which might fortify the blockade of the putative binding site in Ssz1, impairs hand-over of nascent chains to Ssb in yeast.

Thank you for this excellent suggestion. We performed the requested experiments as outlined already above.

Reviewer #3 (Remarks to the Author):

This is an interesting work uncovering the chaperone function of the unusual ribosome bound member of the yeast Hsp70 family, *ssz1* that was previously thought to be acting only as a cochaperone, and studying the dynamics and structural parameters of the nascent polypeptide chain binding by the members of the yeast ribosome associated chaperone and cochaperone triad, RAC-Ssb. The paper is nicely written, experiments are of high quality, and I'd expect this work to make a significant contribution to the field and beyond.

We thank the referee for this positive recognition of our work.

I do however have one serious issue that needs to be addressed at the revision stage. I believe authors should pay more attention to whether or not Ssb binding to the ribosome (both direct and indirect) is actually affected in their experiments, and controls for the Ssb association with the ribosome (rather than only to the nascent polypeptide) should be present. Indeed, for the experiments described on Lines 110-113 and Fig. 1d, authors kind of imply that Ssb binds to the ribosome only through its C-terminus. However, how do authors results fit with the previous observations that Ssb lacking the ribosome binding C-terminal region still binds to the ribosomes (presumably through Zuo1 and Ssz1) and participates in the nascent polypeptide folding? If authors suggest that this binding occurs at a different location, this should be addressed more thoroughly.

We thank the referee for this comment. Indeed, ribosome-binding of Ssb is an important issue in the context of this study.

Two independent studies have addressed ribosome-binding of Ssb thoroughly. The work of the Deuerling/Frydman labs (Hanebuth *et al.* 2016) and our work (Gumiero *et al.* 2016). Both studies fully agree that Ssb mutants lacking the very C-terminus (here collectively termed Ssb- Δ C) no longer directly bind to the ribosome (there is no indication that Ssb- Δ C binds indirectly to ribosomes via RAC). We feel that it would be quite redundant with these published data to again show that Ssb- Δ C does not bind to ribosomes in this work. However, if referee 3 feels it would be helpful, we could right away include such data. Of note, Gumiero *et al.* and Hanebuth *et al.* both come to the conclusion that Ssb- Δ C mutants, even though they do not interact with the ribosome directly, still fully complement growth defects of a Δ *ssb* strain *in vivo*. The unanimous conclusion of both studies is: *direct ribosome-binding is not strictly required for the major chaperone function of Ssb.*

This was a very surprising finding. Hanebuth *et al.* provided evidence that complementation of Δ ssb growth defects by Ssb- Δ C required the functional/physical interaction of Ssb with RAC. Such functional/physical interaction between RAC and Ssb is expected, as RAC is the J-domain partner of Ssb and at least the J-domain interacts with Ssb. To the best of our knowledge additional structural information about direct interactions of RAC and Ssb are not available. Of note, and as outlined above, the direct interaction between RAC and Ssb does not result in detectable recruitment of Ssb- Δ C to ribosomes. Hanebuth *et al.* conclude: "*The reduced binding of Ssb1 Δ 601–13 (i.e. Ssb- Δ C) to nascent chains in the presence of RAC likely reflects the fact that this mutant is no longer able to bind directly to the translation machinery but still interacts with RAC.*"

We would like to conclude: There is no evidence that Ssb- Δ C binds to ribosomes (also not via RAC). However, there is evidence that Ssb- Δ C via RAC participates in the folding of newly synthesized polypeptides.

Even more importantly, for the experiments described in lines 121-126 and shown on Fig. 1, is Ssb indeed bound to the ribosome in these cells? As RAC regulates Ssb binding to the ribosome, it is possible that nascent chains are not transferred from Zuo1 to Ssb simply because Ssb is no longer on the ribosome, rather than because of the absence of Ssz1 contact with Zuo1 per se. It is known that RAC disruption causes a release of ssb from the ribosomes to cytosol.

In such a scenario, interaction between Zuo1 and Ssz1 would of course still play an important role, but in promoting Ssb association with the ribosome, rather than in transferring the nascent polypeptide chain to Ssb (that could be just a consequence of the Ssb/ribosome association). While not invalidating the overall importance of authors' data, such an explanation would significantly change the interpretation, thus I believe it has to be either addressed as an alternative explanation or ruled out by the additional experiments.

We thank the referee for making us look more carefully at Ssb binding in our experimental system!

We previously showed that in total cell extract binding of Ssb to ribosomes is reduced to about 1/3 of the wild type level when RAC is absent or inactive (Gumiero *et al.* 2016). However, we had previously not tested Ssb binding to RNCs in the *in vitro* translation system. We fully agree with the referee: this is specifically important in the experiment shown in Fig. 1e, which addresses the transfer from RAC to Ssb.

We now made up for this flaw and tested ribosome-binding of Ssb in exactly the same experimental set-up as shown in Fig. 1e. The results are presented and described in new Supplementary Fig. 1d. In a nutshell, ribosome-binding of Ssb in the *in vitro* translation system was reduced to approximately 1/3: i) when Zuo1 did not form a complex with Ssz1 (Zuo1 Δ N49), ii) when Ssz1 was absent, or iii) when both Ssz1 and Zuo1 were absent. Thus, loss of ribosome-bound RAC or Ssz1 with respect to ribosome-binding of Ssb exerted the same effect in the crosslinking experiment (Fig. 1e and Supplementary Fig. 1d) and in total cell extract (Gumiero *et al.* 2016). A direct comparison is also possible by viewing Supplementary Fig. 1c (ribosome-binding of Ssb in the presence of the RAC- Δ N49 in total extract) and Supplementary Fig. 1d (ribosome-binding of Ssb in the presence of the RAC- Δ N49 in the *in vitro* translation system).

We conclude: Ribosome-binding of Ssb is reduced to 1/3 when either RAC or Ssz1 is absent from ribosomes. At the same time crosslinking of Ssb to nascent Pgk1 is reduced significantly more severely ($< 1/6$), while crosslinking of Zuo1 to nascent Pgk1 is strongly enhanced (Fig. 1e). We interpret the combined data as strong evidence for an Ssz1-dependent hand-over mechanism to ribosome-bound Ssb.

REVIEWERS' COMMENTS:

Reviewer #1 (Remarks to the Author):

In the revised version of their manuscript "The ribosome-associated complex RAC serves in a relay that directs nascent chains to Ssb" and their "Point-by point answers to reviewer's comments" Zhang et al. have addressed all my concerns satisfactorily. I therefore fully support publication of the revised manuscript in Nature Communications.

Reviewer #2 (Remarks to the Author):

The revised manuscript fully addresses my initial concerns. The new Ssz1 mutant provides convincing evidence that within RAC Ssz1 acts as a molecular chaperone for nascent chains.

Reviewer #3 (Remarks to the Author):

I believe authors addressed my second comment, however I am still somewhat uncertain regarding their response about the ssb-deltaC construct. Authors point to their previous paper showing that Ssb1-deltaC is not binding to the ribosome. However they also agree that ssb1-deltaC compensates for the effects of ssb1/2 deletion, thus retaining its chaperoning function. However the chaperoning function of Ssb is in the folding of nascent polypeptide (according to authors' data as well), and therefore it is hard to imagine how would ssb1-deltaC perform its function in the absence of the interaction with Zuo1/Ssz1 located on the ribosome. I feel that at the very least, discussion with at least hypothetical explanation of this contradiction might be useful. Again, I understand that the previous papers addressed this question, but it seems that they have not really solved it, and as the given paper is specifically addressing interactions between Ssb and RAC, one can't help but ask the question again. Do authors propose that Ssb1-deltaC can achieve its chaperoning role without interaction with Zuo1/Ssz1 complex? Or if does still interact with RAC and nascent polypeptide, why is the result shown on Fig. 1d negative? Also, if authors (as they are implying) do have actual data showing the lack of association of Ssb1-deltaC with the ribosome in the particular experiment shown on Fig. 1d, I'd recommend to include these data, at least in the supplement. Even if this result is not new, it is important control showing that authors' interpretation of this particular experiment is correct.

Otherwise, I think the manuscript is ready to go.

Reply to Referee 3

Reviewer #3 (Remarks to the Author):

I believe authors addressed my second comment, however I am still somewhat uncertain regarding their response about the ssb-deltaC construct. Authors point to their previous paper showing that Ssb1-deltaC is not binding to the ribosome. However they also agree that ssb1-deltaC compensates for the effects of ssb1/2 deletion, thus retaining its chaperoning function. However the chaperoning function of Ssb is in the folding of nascent polypeptide (according to authors' data as well), and therefore it is hard to imagine how would ssb1-deltaC perform its function in the absence of the interaction with Zuo1/Ssz1 located on the ribosome. I feel that at the very least, discussion with at least hypothetical explanation of this contradiction might be useful. Again, I understand that the previous papers addressed this question, but it seems that they have not really solved it, and as the given paper is specifically addressing interactions between Ssb and RAC, one can't help but ask the question again. Do authors propose that Ssb1-deltaC can achieve its chaperoning role without interaction with Zuo1/Ssz1 complex? Or if does still interact with RAC and nascent polypeptide, why is the result shown on Fig. 1d negative? Also, if authors (as they are implying) do have actual data showing the lack of association of Ssb1-deltaC with the ribosome in the particular experiment shown on Fig. 1d, I'd recommend to include these data, at least in the supplement. Even if this result is not new, it is important control showing that authors' interpretation of this particular experiment is correct.

Otherwise, I think the manuscript is ready to go.

We thank the referee for her/his interest in our work and the stimulating discussion! As requested we have included experiments (Supplementary Figs. 1c and 1d), which show that Ssb- Δ C23 is indeed **not** ribosome-associated. We also changed the last paragraph of the Discussion to more clearly address the questions raised by referee 3.

To directly answer the two question addressed by the referee:

- We do **not** think that Ssb functions as a folding helper without its cochaperone RAC (see below).

- The result shown in Fig. 1d analyzes proximity of a **ribosome-bound** nascent chain to Ssb/Ssb- Δ C23 and RAC. The experiment does not analyze proximity of released chains to Ssb/Ssb- Δ C23 or RAC. Ssb- Δ C23 is not ribosome-bound in this experiment (see new Supplementary Fig. 1c and 1d); at the same time Ssb- Δ C23 is not efficiently crosslinked to the ribosome-bound nascent chain. The important message of Fig. 1d in the context of this study is: **when Ssb- Δ C23 does not bind to ribosomes, it also does not efficiently bind to ribosome-bound nascent chains. In this specific situation RAC retains contact with the growing nascent chain.**

Some more detailed comments:

Stable ribosome-binding of Ssb is not essential for its *de novo* protein folding function *in vivo*. Two independent studies, one from our labs (Rospert/Sinning) ¹ and one from the Deuring/Frydman ² groups came to the exactly same conclusion. As suggested by the referee we now included additional data, which confirm these earlier observations: see Supplementary Fig. 1c (ribosome-binding of Ssb- Δ C23 in total extract) and Supplementary Fig. 1d (ribosome-binding of Ssb- Δ C23 under the very same conditions as applied in the crosslinking experiments, as e.g. shown in Fig. 1d).

Now, how does cytosolic Ssb- Δ C23 perform its function in protein folding, and how may RAC contribute to this?

So far no other J-domain protein was found to act as a cochaperone of Ssb. We thus assume that i) RAC acts also as a cochaperone for Ssb- Δ C23, and ii) there must be a cytosolic pool of RAC. To the best of our knowledge this has not been firmly demonstrated: in cell extract RAC is fully ribosome-bound (see e.g. this work, Supplementary Fig. 1d). However, several lines of evidence suggest that binding of RAC to ribosomes is transient. First, the relative concentration of ribosomes:Ssb:RAC in wild type cells is approximately 1:1:0.3³. These numbers already suggest that RAC is not permanently fixed to a single ribosome *in vivo*. Moreover, it was shown that the concentration of RAC (but not of Ssb!) can be severely down-regulated to about 1-2% of the wild type level, without negative effects on cell growth⁴. Thus, about 3 molecules of RAC per 1000 ribosomes are sufficient for normal cell growth. This strongly suggests a fast on/off rate of RAC with respect to ribosome binding in living cells. Please also note, that in cell extract about 40-50% of Ssb is localized to the cytosol. This cytosolic pool of Ssb forms complexes with kinases and phosphates and is involved in the regulation of signaling pathways. This function of Ssb is not directly connected to protein folding. It was shown that RAC is required for the cytosolic signaling function of Ssb, indicating that there must be a cytosolic pool of RAC^{5,6}.

We would also like to draw the attention of the referee to other eukaryotic cells. Mammalian cells do not possess an Hsp70 homolog, which interacts with ribosomes directly⁷. However, mammalian cells possess a ribosome-bound RAC homolog, termed mRAC^{7,8}. Based on our previous findings we suggested a model for mammalian cotranslational protein folding in that mRAC directly bind to nascent chains, either by its MPP11 or Hsp70L1 subunit, and hands over these nascent chains specifically to Hsp70 (encoded by HSPA1A and HSPA1B), which is localized to the cytosol. In our opinion the situation in mammalian cells resembles the situation of yeast strains expressing Ssb- Δ C23.

Based on the currently available data we speculate that in situations, in which transfer of nascent chains from RAC to Ssb cannot occur directly at the tunnel exit, RAC leaves the ribosome in a transient complex with the nascent chain, from which the newly synthesized protein is transferred to Ssb/Ssb- Δ C23 in the cytosol or to Ssa (like for mRAC). The model will be experimentally tested in future studies.

References

1. Gumiero, A. *et al.* Interaction of the cotranslational Hsp70 Ssb with ribosomal proteins and rRNA depends on its lid domain. *Nat. Commun.* **7**, 1-12 (2016).
2. Hanebuth, M. A. *et al.* Multivalent contacts of the Hsp70 Ssb contribute to its architecture on ribosomes and nascent chain interaction. *Nat. Commun.* **7**, 13695 (2016).
3. Raue, U., Oellerer, S. & Rospert, S. Association of protein biogenesis factors at the yeast ribosomal tunnel exit is affected by the translational status and nascent polypeptide sequence. *J. Biol. Chem.* **282**, 7809-7816 (2007).
4. Hundley, H. *et al.* The *in vivo* function of the ribosome-associated Hsp70, Ssz1, does not require its putative peptide-binding domain. *Proc. Natl. Acad. Sci. U S A* **99**, 4203-4208 (2002).
5. Hübscher, V. *et al.* The Hsp70 homolog Ssb and the 14-3-3 protein Bmh1 jointly regulate transcription of glucose repressed genes in *Saccharomyces cerevisiae*. *Nucleic Acids Res* **44**, 5629-5645 (2016).
6. Mudholkar, K., Fitzke, E., Prinz, C., Mayer, M. P. & Rospert, S. The Hsp70 homolog Ssb affects ribosome biogenesis via the TORC1-Sch9 signaling pathway. *Nat Commun* **8**, 1-14 (2017).

7. Jaiswal, H. *et al.* The chaperone network connected to human ribosome-associated complex (mRAC). *Mol. Cell. Biol.* **31**, 1160-1173 (2011).
8. Otto, H. *et al.* The chaperones MPP11 and Hsp70L1 form the mammalian ribosome-associated complex. *Proc. Natl. Acad. Sci. USA* **102**, 10064-10069 (2005).